# Overcoming Output Dimension Collapse: When Sparsity Enables Zero-shot Brain-to-image Reconstruction at Small Data Scales

**Kenya Otsuka**                                                  *otsuka.kenya.54m@st.kyoto-u.ac.jp*
*Graduate School of Informatics, Kyoto University*
*Computational Neuroscience Laboratories, Advanced Telecommunications Research Institute International*

**Yoshihiro Nagano**                                                  *nagano@i.kyoto-u.ac.jp*
*Graduate School of Informatics, Kyoto University*
*Computational Neuroscience Laboratories, Advanced Telecommunications Research Institute International*

**Yukiyasu Kamitani**                                                  *kamitani@i.kyoto-u.ac.jp*
*Graduate School of Informatics, Kyoto University*
*Computational Neuroscience Laboratories, Advanced Telecommunications Research Institute International*
*Guardian Robot Project, RIKEN*

**Reviewed on OpenReview:** *https://openreview.net/forum?id=RQiXUK4kQr*

## Abstract

Advances in brain-to-image reconstruction are enabling us to externalize the subjective visual experiences encoded in the brain as images. A key challenge in this task is data scarcity: a translator that maps brain activity to latent image features is trained on a limited number of brain-image pairs, making the translator a bottleneck for zero-shot reconstruction beyond the training stimuli. In this paper, we mathematically analyze the behavior of two translators commonly used in recent reconstruction pipelines: naive multivariate linear regression and sparse multivariate linear regression. We define the data scale as the ratio of the number of training samples to the latent feature dimensionality and characterize the behavior of each model across data scales. Building on a standard structural property of naive multivariate regression, we first show that the resulting "output dimension collapse" can become a practical generalization bottleneck in brain-to-image reconstruction. We introduce the best prediction diagnostic, which is computable without brain activity, to quantify the practical impact of this collapse. We then analyze sparse linear regression models in a student–teacher framework and derive expressions for the prediction error in terms of data scale and other sparsity-related parameters. Our analysis clarifies when variable selection can reduce prediction error at small data scales by exploiting the sparsity of the brain-to-feature mapping. Our findings provide quantitative guidelines for diagnosing output dimension collapse and for designing effective translators and feature representations for zero-shot reconstruction.

## 1  Introduction

Advances in brain-to-image reconstruction are enabling us to externalize the subjective visual experiences encoded in the brain as images. To uncover neural representations and move toward practical applications in medicine and industry, we need reconstruction methods that generalize to a broad range of subjective visual experiences. A key challenge in this task is data scarcity: the space of possible visual stimuli is vast, yet we can only collect a limited number of brain-image pairs for training. To capture the full spectrum of subjective experiences, a model must be able to predict previously unseen stimuli, i.e., zero-shot prediction (Shirakawa et al., 2025). Despite its importance, we still lack guidelines for achieving accurate reconstruction under these constraints.

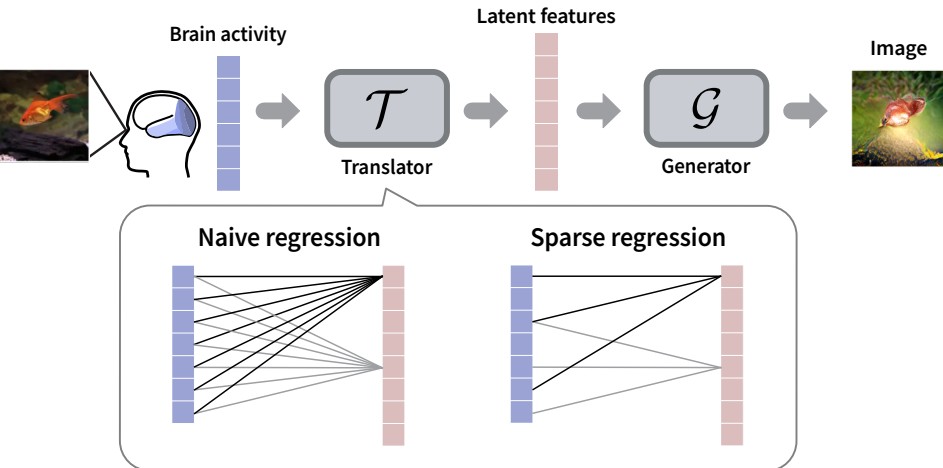

Figure 1: **Translator–Generator pipeline.** The translator converts brain activity into latent features, and the generator transforms latent features into reconstructed images. A naive multivariate linear regression model and a sparse multivariate linear regression model are commonly used as translators. A naive model uses a shared set of input variables to predict all output dimensions, and a sparse model performs variable selection for each output dimension.

Many recent reconstruction methods employ a typical architecture, referred to as the Translator–Generator pipeline (Shirakawa et al., 2025) (Fig. 1). In this framework, the translator converts brain activity into latent features that represent the visual content, and the generator transforms these features into reconstructed images. Latent features are typically high-dimensional to represent a broad spectrum of visual stimuli. Recent pipelines often leverage powerful generators, but the translator must be trained using only a limited number of brain-feature pairs. We quantify this data limitation using the data scale $n/d_{\text{out}}$, where $n$ is the number of training pairs and $d_{\text{out}}$ is the latent feature dimensionality. The translator becomes a key bottleneck for zero-shot reconstruction because it must predict valid latent features for unseen stimuli when $n/d_{\text{out}} < 1$.

This paper focuses on the translator in reconstruction pipelines, with a particular focus on linear translators. While several reconstruction methods have adopted nonlinear translators, linear translators are also used in a range of reconstruction studies. A common linear translator design is what we call naive multivariate linear regression, which uses a shared set of input variables to predict all output dimensions (e.g., ridge regression). This design is straightforward and low-cost, making it a common choice in recent studies (Seeliger et al., 2018; Ozcelik & VanRullen, 2023; Takagi & Nishimoto, 2023). Another approach is sparse multivariate linear regression, which performs variable selection for each output dimension (e.g., Lasso, ARD, filter methods). Earlier work adopted sparse regression as the translator, assuming that "brain-like" latent features induce a sparse brain-to-feature mapping (Miyawaki et al., 2008; Zhang et al., 2018; Shen et al., 2019). Theoretical results suggest that sparse regression can be sample-efficient when the underlying mapping is sparse (Donoho, 2006; Wainwright, 2009b). Notably, these linear translators are not only classical baselines but also central components in many recent reconstruction pipelines.

In brain-to-image reconstruction, Shirakawa et al. (2025) pointed out a phenomenon in naive multivariate linear translators called "output dimension collapse" (ODC). They reported that several high-profile reconstruction methods fail to generalize beyond the training domain. They attributed this failure to ODC, in which predictions are confined to the span of the training outputs in latent feature space under dataset bias. ODC is a challenge that should be addressed to achieve zero-shot prediction, but we still lack insight into when it occurs and how it can be overcome.

In this paper, we analyze translator prediction under small data scales from a mathematical perspective. First, complementing prior work that emphasizes dataset bias (Shirakawa et al., 2025), we show that small data scales can cause ODC in naive multivariate linear regression. Using the mathematically derived best pre-

diction under this collapse, we evaluate its impact on reconstruction pipelines that employ naive translators. Second, we examine when sparsity can enable accurate prediction at small data scales by mathematically characterizing the prediction error of naive and sparse translators. The main contributions of this work are threefold: (1) a formalization and analytic diagnosis of data-scale-induced ODC for naive linear translators in brain-to-image reconstruction, based on the best prediction under collapse; (2) a quantitative error theory for naive and sparse translators across data scales; and (3) a characterization of when sparsity can improve prediction at small data scales.

## 2 Preliminaries

Many recent reconstruction methods employ a typical architecture, referred to as the Translator–Generator pipeline (Shirakawa et al., 2025) (Fig. 1). In this framework, a module called the translator converts brain activity into latent features, and the generator transforms these features into reconstructed images. The translator is trained on paired brain activity and latent feature data. The generator is typically a pre-trained or pre-defined model that generates images from the predicted features. Latent features serve as an intermediate representation that links neural signals and visual content, and recent studies often utilize artificial neural network features, such as Convolutional Neural Networks (CNNs) and Contrastive Language-Image Pretraining (CLIP).

In this study, we focus on the translator module, which maps brain activity $\boldsymbol{x} \in \mathbb{R}^{d_{\text{in}}}$ to latent features $\boldsymbol{y} \in \mathbb{R}^{d_{\text{out}}}$. Our analysis focuses on linear translators, while noting that some reconstruction methods employ nonlinear ones. The training set consists of $n$ paired samples of brain activity and latent features, denoted by $\mathcal{D} = \{(\boldsymbol{x}^{(i)}, \boldsymbol{y}^{(i)})\}_{i=1}^{n}$. The latent features $\boldsymbol{y}$ are usually high-dimensional representations because they represent a broad spectrum of visual stimuli. Meanwhile, the number of available brain activity samples $n$ is limited; $n$ is much smaller than $d_{\text{out}}$. We therefore define the data scale as the ratio $n/d_{\text{out}}$, which is typically much smaller than 1 in reconstruction studies. To achieve zero-shot reconstruction, the translator must predict valid latent features for unseen stimuli, even under small data scales.

A common approach to implementing a translator is naive multivariate linear regression, which utilizes a single, shared set of input variables to predict all output dimensions. Representative methods include ridge regression, kernel linear regression, and partial least squares (PLS). These methods differ in how they regularize or project the data, but still rely on shared inputs for all output dimensions.

Another approach is sparse multivariate linear regression, which performs variable selection for each output dimension. According to the classic taxonomy (Guyon & Elisseeff, 2003), selection strategies fall into three groups: wrapper methods (Kohavi & John, 1997), which iteratively search feature subsets by maximizing predictive performance; embedded methods (e.g., Lasso (Tibshirani, 1996), Automatic Relevance Determination (ARD) (MacKay, 1992)), which embed sparsity directly in the training objective; and filter methods (e.g., CFS (Hall, 2000), SIS (Fan & Lv, 2008)), which rank inputs independently of the learner using simple scores (e.g., correlation, mutual information). In high-dimensional settings such as brain-to-image reconstruction, computational speed is essential, and wrapper methods are typically the most computationally expensive, embedded methods have intermediate cost, and filter methods are usually the least expensive.

## 3 Related work

Some recent reconstruction works rely on the naive linear regression model (Seeliger et al., 2018; Ozcelik & VanRullen, 2023; Takagi & Nishimoto, 2023). Although it is attractive due to its simplicity and computational efficiency, Shirakawa et al. (2025) pointed out its geometric limitation as output dimension collapse (ODC). Shirakawa et al. (2025) showed that such a collapse occurs when the training dataset lacks sufficient diversity and the latent features exhibit a cluster structure. In such scenarios, the predicted outputs are pulled toward the clusters in the training dataset, making generalization to unseen stimuli difficult.

In contrast to naive translators, several reconstruction studies have adopted sparse regression as the translator (Miyawaki et al., 2008; Zhang et al., 2018; Shen et al., 2019). In brain-to-image reconstruction, using "brain-like" latent features makes it reasonable to assume a sparse brain-to-feature mapping. Neurophysiology

shows that the visual cortex exhibits local or selective activity in response to visual stimuli (Hubel & Wiesel, 1962; De Valois et al., 1982; Vinje & Gallant, 2000). Correspondingly, several reconstruction studies have adopted latent representations that exhibit similar locality and selectivity, such as local bases or intermediate CNN features (Miyawaki et al., 2008; Zhang et al., 2018; Shen et al., 2019). Under these "brain-like" features, only a small subset of voxels should influence each latent dimension.

Previous theoretical work demonstrates that sparsity is particularly beneficial when the sample size is smaller than the dimensionality of the data. We call an input–output mapping $s$-sparse if, for each output coordinate, the corresponding weight vector has at most $s$ non-zero entries (i.e., the output depends on only $s$ of the $d_{in}$ input variables). For a truly $s$-sparse input-output mapping, information-theoretic bounds show that an ideal estimator can attain near-oracle prediction error once $n \gtrsim s \log(d_{in}/s)$ (Wainwright, 2009a). The Lasso is the most thoroughly analyzed among practical sparse methods, and theory confirms it reaches essentially the same rate and achieves accurate prediction with $n = \mathcal{O}(s \log(d_{in}))$ (Donoho, 2006; Wainwright, 2009b). The SIS filter method is also proven to recover the correct support of the mapping with high probability when $d_{in}$ grows exponentially in $n$ (Fan & Lv, 2008).

## 4 Output dimension collapse by small data scales

Although both naive and sparse linear models are used as the translator module, several recent works rely on the naive model for simplicity and computational efficiency (Seeliger et al., 2018; Ozcelik & VanRullen, 2023; Takagi & Nishimoto, 2023). This section focuses on the naive translator and clarifies how a standard geometric property of the naive model can become a bottleneck in reconstruction pipelines at small data scales.

Output dimension collapse (ODC) refers to a phenomenon in naive multivariate linear regression: its predictions are confined to a low-dimensional subspace spanned by the training outputs in the latent feature space. This restriction can be especially critical to zero-shot prediction when the training-output subspace is low-dimensional, because unfamiliar features are inevitably compressed into that subspace. To understand why this collapse occurs, let us examine the predicted features of ridge regression. In ridge regression, the fitted weight matrix $\hat{W} \in \mathbb{R}^{d_{in} \times d_{out}}$ admits the following closed-form solution:

$$\hat{W} = \underset{W}{\operatorname{argmin}} \sum_{i=1}^{n} \|\boldsymbol{y}^{(i)} - W^\top \boldsymbol{x}^{(i)}\|^2 + \lambda \|W\|_F^2 \tag{1}$$

$$= \left(X_{tr}^\top X_{tr} + \lambda I\right)^{-1} X_{tr}^\top Y_{tr}, \tag{2}$$

where $X_{tr} = [\boldsymbol{x}^{(1)}, \boldsymbol{x}^{(2)}, \ldots, \boldsymbol{x}^{(n)}]^\top \in \mathbb{R}^{n \times d_{in}}$ and $Y_{tr} = [\boldsymbol{y}^{(1)}, \boldsymbol{y}^{(2)}, \ldots, \boldsymbol{y}^{(n)}]^\top \in \mathbb{R}^{n \times d_{out}}$ stack the training inputs and outputs row-wise. For a test input brain activity $\boldsymbol{x}_{te}$, the prediction is

$$\hat{\boldsymbol{y}}_{te} = \hat{W}^\top \boldsymbol{x}_{te} \tag{3}$$

$$= Y_{tr}^\top X_{tr} \left(X_{tr}^\top X_{tr} + \lambda I\right)^{-1} \boldsymbol{x}_{te} \tag{4}$$

$$= Y_{tr}^\top \boldsymbol{m} = \sum_{i=1}^{n} m_i \boldsymbol{y}^{(i)} \tag{5}$$

where $\boldsymbol{m} = X_{tr} \left(X_{tr}^\top X_{tr} + \lambda I\right)^{-1} \boldsymbol{x}_{te} \in \mathbb{R}^n$, and $m_i \in \mathbb{R}$ is the $i$-th element of $\boldsymbol{m}$. Thus, the prediction $\hat{\boldsymbol{y}}_{te}$ lies in the linear span of the training outputs $\{\boldsymbol{y}^{(i)}\}_{i=1}^{n}$ for any test input $\boldsymbol{x}_{te}$ (Fig. 2A). This span is determined solely by the training outputs and does not depend on the input brain data; given the same training outputs, the same property holds regardless of measurement modality (e.g., fMRI, MEG), subject, or decoding task (e.g., perception, imagery, or illusion). This property is not limited to ridge regression; it also holds for other naive multivariate linear regression methods, such as kernel linear regression and PLS (see Appendix B.1 for details). This property follows directly from the closed-form ridge solution, but it can become a major limitation in brain-to-image reconstruction.

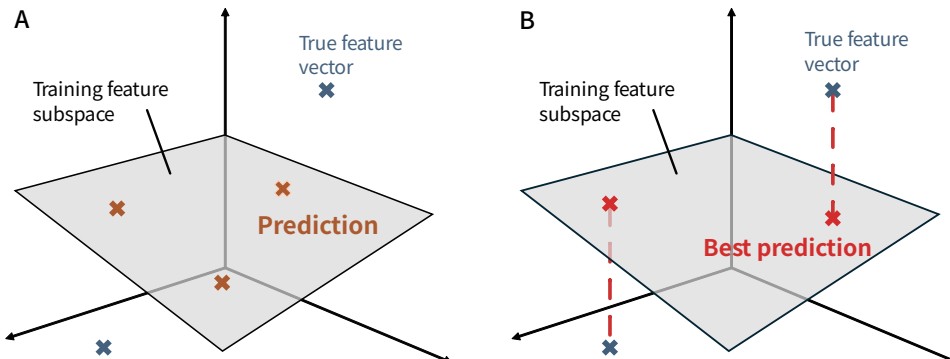

Figure 2: **Output dimension collapse.** (A) The predictions become restricted to a low-dimensional subspace determined by the training outputs. (B) The best prediction within the training feature subspace provides the lower bound for prediction error in naive multivariate linear regression.

ODC becomes especially pronounced when the training outputs occupy a low-dimensional subspace in latent feature space. Shirakawa et al. (2025) have emphasized dataset-bias scenarios as one mechanism that can induce such a restriction. Here, we focus on a complementary but equally critical factor: small data scales. Even in the absence of explicit cluster structure, having $n < d_{\text{out}}$ inherently forces all predictions into an at most $n$-dimensional subspace. As $n/d_{\text{out}}$ grows smaller, this low-dimensional subspace becomes more limiting, producing severe ODC. Consequently, regardless of whether the training outputs form visible clusters, small data scales alone can lead to the same fundamental difficulty in zero-shot prediction.

To formally characterize this collapse, we derive the best prediction attainable within the collapsed subspace. This best prediction establishes a lower bound for prediction error under ODC. Let $\mathcal{A} = \left\{ Y_{\text{tr}}^{\top} \boldsymbol{m} \mid \boldsymbol{m} \in \mathbb{R}^{n} \right\}$ denote the subspace defined by linear combinations of the training outputs $\left\{ \boldsymbol{y}^{(i)} \right\}_{i=1}^{n}$. The vector $\hat{\boldsymbol{y}}_{\text{te}}^{\text{opt}} \in \mathcal{A}$ that is closest to the true latent feature vector $\boldsymbol{y}_{\text{te}}$ can be derived analytically as:

$$\hat{\boldsymbol{y}}_{\text{te}}^{\text{opt}} = \underset{\hat{\boldsymbol{y}}_{\text{te}} \in \mathcal{A}}{\operatorname{argmin}} \| \hat{\boldsymbol{y}}_{\text{te}} - \boldsymbol{y}_{\text{te}} \|^{2} \tag{6}$$

$$= Y_{\text{tr}}^{\top} \left( Y_{\text{tr}} Y_{\text{tr}}^{\top} \right)^{\dagger} Y_{\text{tr}} \boldsymbol{y}_{\text{te}} \tag{7}$$

where $A^{\dagger}$ denotes the Moore-Penrose pseudo-inverse of $A$ (see Appendix B.2 for details). This expression shows that $\hat{\boldsymbol{y}}_{\text{te}}^{\text{opt}}$ is the orthogonal projection of the true latent feature vector onto the subspace $\mathcal{A}$ (Fig. 2B). Because it represents the best achievable prediction within this subspace, computing it in a given reconstruction pipeline allows us to assess how strongly ODC constrains its predictions directly. A practical recipe for computing the best prediction $\hat{\boldsymbol{y}}_{\text{te}}^{\text{opt}}$ is provided in Appendix B.3.

The best prediction $\hat{\boldsymbol{y}}_{\text{te}}^{\text{opt}}$ provides the lower bound for prediction error in naive multivariate linear regression:

$$\forall \hat{\boldsymbol{y}}_{\text{te}} \in \mathcal{A}, \quad \| \hat{\boldsymbol{y}}_{\text{te}}^{\text{opt}} - \boldsymbol{y}_{\text{te}} \|^{2} \leq \| \hat{\boldsymbol{y}}_{\text{te}} - \boldsymbol{y}_{\text{te}} \|^{2}. \tag{8}$$

This inequality shows that no prediction $\hat{\boldsymbol{y}}_{\text{te}}$ can ever outperform the best prediction $\hat{\boldsymbol{y}}_{\text{te}}^{\text{opt}}$. Moreover, the prediction error $\| \hat{\boldsymbol{y}}_{\text{te}} - \boldsymbol{y}_{\text{te}} \|^{2}$ can be decomposed into two components:

$$\| \hat{\boldsymbol{y}}_{\text{te}} - \boldsymbol{y}_{\text{te}} \|^{2} = \| \hat{\boldsymbol{y}}_{\text{te}} - \hat{\boldsymbol{y}}_{\text{te}}^{\text{opt}} \|^{2} + \| \hat{\boldsymbol{y}}_{\text{te}}^{\text{opt}} - \boldsymbol{y}_{\text{te}} \|^{2}. \tag{9}$$

The first term $\| \hat{\boldsymbol{y}}_{\text{te}} - \hat{\boldsymbol{y}}_{\text{te}}^{\text{opt}} \|^{2}$ corresponds to the within-subspace error, and the second term $\| \hat{\boldsymbol{y}}_{\text{te}}^{\text{opt}} - \boldsymbol{y}_{\text{te}} \|^{2}$ corresponds to the out-of-subspace error. When $n < d_{\text{out}}$, the out-of-subspace error is generally non-zero, implying an irreducible error dictated by the training outputs. This error originates from the training outputs being insufficient to cover the universe of possible images. We later verify empirically that this irreducible error is large enough to degrade the quality of reconstructed images, and most of the observed prediction error can be explained by this irreducible out-of-subspace component.

## 5 Empirical study: ODC on real data

We showed that naive multivariate linear regression is prone to output dimension collapse at small data scales. We next examine how this collapse affects brain-to-image reconstruction in practice. Specifically, we first analyze the best prediction $\hat{\boldsymbol{y}}_{\text{te}}^{\text{opt}}$ under various data scales and then compare $\hat{\boldsymbol{y}}_{\text{te}}^{\text{opt}}$ with the actual prediction from brain $\hat{\boldsymbol{y}}_{\text{te}}$.

Our experiments mainly follow the method of Shen et al. (2019), which employs a Translator–Generator framework to reconstruct images from brain activity (see also Appendix B.4). To observe the effect of ODC, we replace their sparse translator with a naive multivariate linear regression model. For the generator, we mainly adopt Deep Image Prior (DIP) (Ulyanov et al., 2018) as the image prior to visualize the latent features with weak prior influence, following the protocol of Shen et al. (2019) for iterative image optimization. We also evaluated the method of Ozcelik and VanRullen (Ozcelik & VanRullen, 2023) as an additional baseline, which adopts naive multivariate linear regression as the translator. We selected the Deeprecon dataset (Shen et al., 2019), designed explicitly for brain-to-image reconstruction. The training set comprises 1,200 natural images from ImageNet (Deng et al., 2009), and the test set consists of 50 natural images from different categories (unseen during training) and 40 artificial shapes drawn entirely from outside the training domain. Zero-shot prediction beyond the training stimuli is crucial in brain-to-image reconstruction research, so we chose a dataset whose training and test partitions are strictly separated, enabling rigorous zero-shot evaluation.

To see how the data scale drives ODC, we progressively reduce the number of training samples from 1,200 down to 600, 300, and 150, and we compute the best prediction $\hat{\boldsymbol{y}}_{\text{te}}^{\text{opt}}$ in each condition. Fig. 3A plots the best prediction error $\|\hat{\boldsymbol{y}}_{\text{te}}^{\text{opt}} - \boldsymbol{y}_{\text{te}}\|^2$ for each test sample across these different values of $n$. As $n$ decreases, the error grows, indicating that the dimension of the training subspace is shrinking and cannot fully capture the variability in the true latent features. Fig. 3B shows example reconstructions generated from the true features $\mathcal{G}(\boldsymbol{y}_{\text{te}})$ and the best prediction $\mathcal{G}(\hat{\boldsymbol{y}}_{\text{te}}^{\text{opt}})$ at each data scale. While the reconstruction from the uncollapsed true feature $\mathcal{G}(\boldsymbol{y}_{\text{te}})$ is nearly perfect, the quality of the reconstruction from the best prediction $\mathcal{G}(\hat{\boldsymbol{y}}_{\text{te}}^{\text{opt}})$ worsens noticeably as $n$ decreases. To disentangle data-scale effects from dataset bias reported by Shirakawa et al. (2025), we additionally evaluate a setting where train and test categories overlap, and confirm the same trend holds: the inevitable (out-of-subspace) error remains large (Fig. A1). These results suggest that in reconstruction methods using naive multivariate linear regression, the data scale critically affects reconstruction quality through ODC.

To assess how ODC limits actual prediction performance, we use the full set of 1,200 training images and compare the best prediction $\hat{\boldsymbol{y}}_{\text{te}}^{\text{opt}}$ with the prediction from brain activity $\hat{\boldsymbol{y}}_{\text{te}}$. Fig. 4A compares the prediction errors of the best prediction $\hat{\boldsymbol{y}}_{\text{te}}^{\text{opt}}$ and the brain prediction $\hat{\boldsymbol{y}}_{\text{te}}$. For natural images, the best prediction error accounts for 70% of the brain prediction error, and even for artificial images, the proportion reaches 40% (left panel). Examining individual samples reveals a strong correlation between the best and brain prediction errors (right panel). Samples that lie farther from the subspace (i.e., with larger best prediction errors) exhibit correspondingly larger total prediction errors. These observations show that a substantial portion of the prediction error corresponds to the irreducible out-of-subspace error, i.e., a direct consequence of ODC. Fig. A2 presents the corresponding results with subsampled training sets. Fig. 4B shows the reconstructed images generated from the best prediction $\mathcal{G}(\hat{\boldsymbol{y}}_{\text{te}}^{\text{opt}})$ and the brain prediction $\mathcal{G}(\hat{\boldsymbol{y}}_{\text{te}})$. Both reconstructions are similarly blurred, markedly different from the "true" reconstruction. These trends are consistent across subjects (Fig. A3, Fig. A4) and remain qualitatively similar under stronger generator choices (Fig. A5). We also observe similar results with another reconstruction method (Ozcelik & VanRullen, 2023) (Fig. A6). As a reference point, Shen et al.'s original method (Shen et al., 2019) makes different design choices (e.g., a sparse translator) and can yield sharper images.

These results confirm that naive multivariate linear regression suffers severely from ODC due to small data scales. Even the best possible subspace prediction $\hat{\boldsymbol{y}}_{\text{te}}^{\text{opt}}$ deviates substantially from the true latent $\boldsymbol{y}_{\text{te}}$, creating an irreducible error that impedes zero-shot reconstruction. In turn, actual predictions $\hat{\boldsymbol{y}}_{\text{te}}$ inherit this limitation and produce blurred reconstructions. These results underscore that naive regression is poorly suited to the fundamental structure of reconstruction datasets: the vast image space and the limited training data.

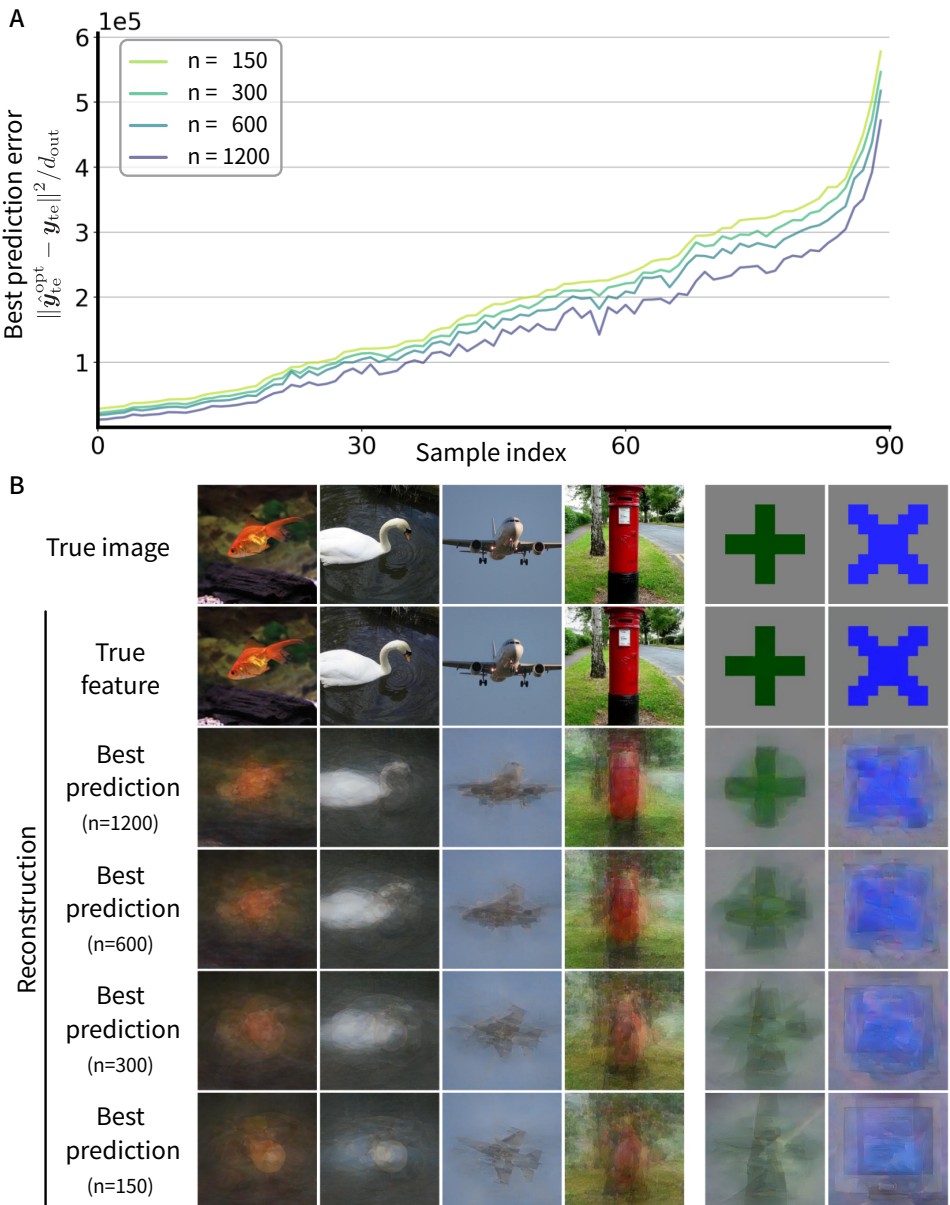

Figure 3: **The best predictions for different training data sizes.** (A) The best prediction error $\frac{1}{d_{\mathrm{out}}}\|\hat{\boldsymbol{y}}_{\mathrm{te}}^{\mathrm{opt}} - \boldsymbol{y}_{\mathrm{te}}\|^2$ for each test sample across training data sizes 1200, 600, 300, and 150. The vertical axis represents the error of the best prediction, and the horizontal axis represents the sample index sorted by the error at $n = 150$. (B) Representative reconstructions from the best predictions for different training data sizes. From top: ground truth, reconstructed images generated from the true features $\mathcal{G}(\boldsymbol{y}_{\mathrm{te}})$, the best predictions $\mathcal{G}(\hat{\boldsymbol{y}}_{\mathrm{te}}^{\mathrm{opt}})$ using 1200, 600, 300, and 150 samples. The left four columns show natural images, and the right two columns show artificial shape images.

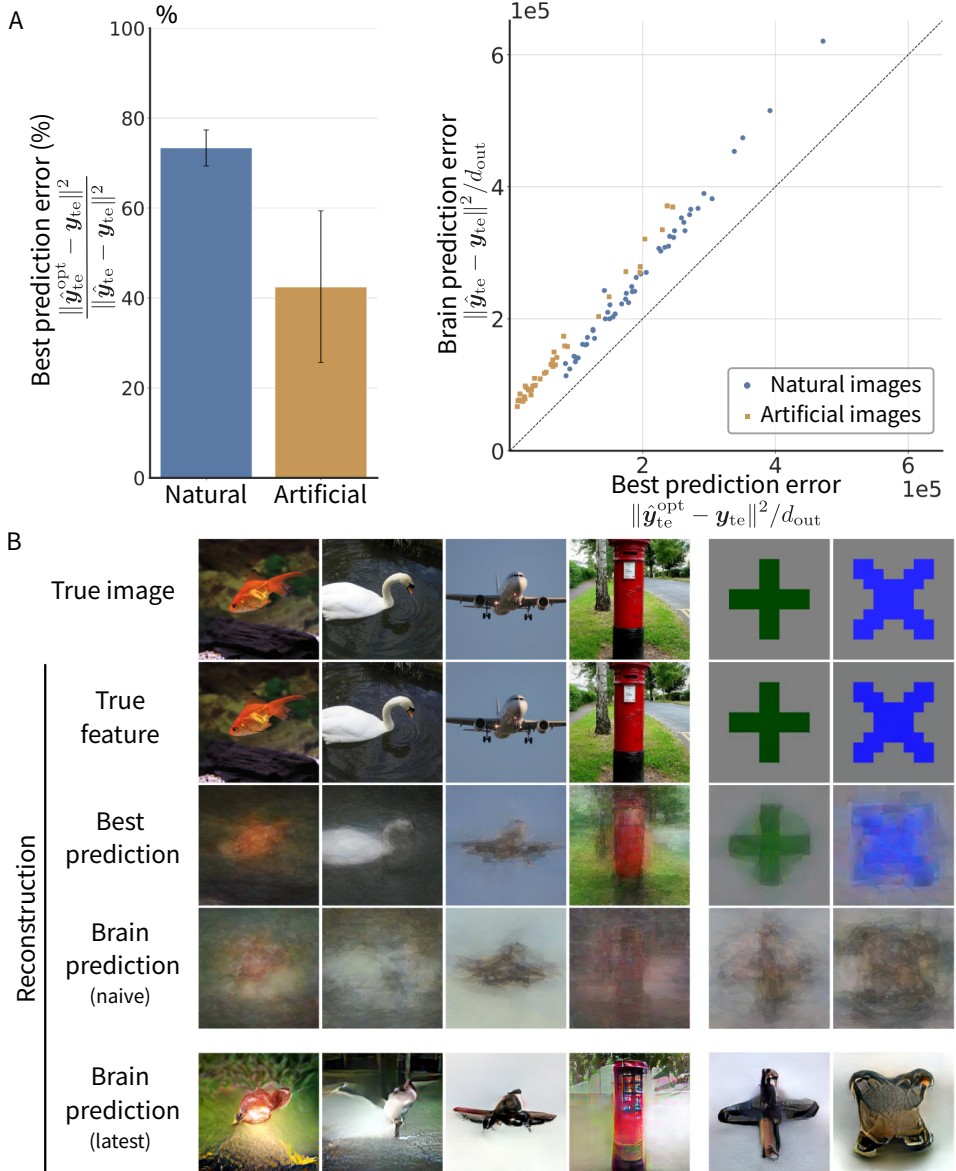

Figure 4: **Comparison of the best prediction and the brain prediction.** (A) Left: Percentage of the best prediction error relative to the brain prediction error $\|\hat{\boldsymbol{y}}_{te}^{opt} - \boldsymbol{y}_{te}\|^2 / \|\hat{\boldsymbol{y}}_{te} - \boldsymbol{y}_{te}\|^2$. Blue for natural images, orange for artificial shapes. The error bars denote the standard deviation across samples. Right: Sample-wise comparison of best prediction error $\frac{1}{d_{out}}\|\hat{\boldsymbol{y}}_{te}^{opt} - \boldsymbol{y}_{te}\|^2$ and brain prediction error $\frac{1}{d_{out}}\|\hat{\boldsymbol{y}}_{te} - \boldsymbol{y}_{te}\|^2$. Each dot represents one image, and the dotted line indicates where the two errors are equal. (B) Representative reconstructions generated from the best predictions and brain predictions. From top: ground truth, reconstructions from the true features $\mathcal{G}(\boldsymbol{y}_{te})$, the best predictions $\mathcal{G}(\hat{\boldsymbol{y}}_{te}^{opt})$, brain predictions $\mathcal{G}(\hat{\boldsymbol{y}}_{te})$. The bottom row shows reconstructions from the original method of Shen et al. (2019). The left four columns show natural images, and the right two columns show artificial shape images.

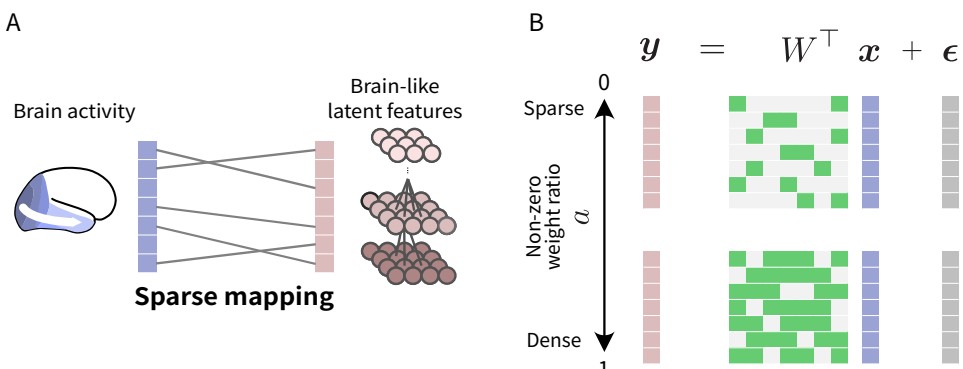

Figure 5: **Sparse brain-to-feature mapping.** (A) If the brain and the latent features exhibit local or selective activity, their mapping should become sparse. (B) Teacher model in our student–teacher framework, reflecting the sparse structure of the brain-to-feature mapping. The smaller the non-zero weight ratio $a$, the sparser the input-output mapping is.

## 6  Leveraging sparse brain-to-feature structure

We have shown that naive multivariate linear regression suffers from output dimension collapse at small data scales, reflecting the high-dimensional, sample-limited structure of reconstruction datasets. In contrast, with "brain-like" latent representations that reflect locality or selectivity in visual cortical responses, it is natural to model the brain-to-feature mapping as sparse: each output feature depends on only a small subset of voxels (Fig. 5A). Sparse regression provides an inductive bias that matches this structure by selecting a subset of inputs. We therefore theoretically analyze sparse regression to clarify how exploiting sparsity affects prediction. We first show how sparsity changes the subspace constraint underlying ODC, and we then analyze prediction error in a student–teacher setting for a broad class of sparse regression procedures.

Importantly, sparse models are not forced into ODC at small data scales ($n < d_{\text{out}}$). As shown previously, sharing a single set of input variables across all output dimensions forces any prediction to be a linear combination of the training outputs, $\hat{\boldsymbol{y}}_{\text{te}} = Y_{\text{tr}}^\top \boldsymbol{m}$ for some $\boldsymbol{m} \in \mathbb{R}^n$, so all predictions lie in their span and $\text{rank}(\hat{W}) \leq n$. In contrast, a sparse model can select different subsets of input variables for different output dimensions, so its predictions are not confined to the training-output subspace; in particular, $\text{rank}(\hat{W})$ can exceed $n$. Thus, sparse regression can represent predictions beyond the training-output subspace. Conceptually, ODC results from a mismatch between the implicit parameter reduction of the naive model and the data structure. When the true mapping is sparse, explicit reduction through variable selection that matches this structure should be preferred.

To clarify how the translator behaves under sparsity, we introduce a student–teacher framework. For simplicity, we assume $d := d_{\text{in}} = d_{\text{out}}$; $\boldsymbol{x} \in \mathbb{R}^{d_{\text{in}}}$ and $\boldsymbol{y} \in \mathbb{R}^{d_{\text{out}}}$ share the same dimensionality, and we denote this common dimension by $d$. Specifically, we consider a sparse data-generating process (Assumption 1) as a teacher model together with a generic sparse regression procedure (Definition 1) as a student model.

**Assumption 1** (Sparse linear data-generating process). *Each sample pair $\left(\boldsymbol{x}^{(i)}, \boldsymbol{y}^{(i)}\right)$ is generated by a linear teacher as*

$$\boldsymbol{y}^{(i)} = W^\top \boldsymbol{x}^{(i)} + \boldsymbol{\epsilon}^{(i)}, \tag{10}$$

*where the brain activity $\boldsymbol{x}^{(i)}$ is generated by sampling $\boldsymbol{x}^{(i)} \sim \mathcal{N}(\boldsymbol{0}, I_d)$, the noise $\boldsymbol{\epsilon}^{(i)} \sim \mathcal{N}(\boldsymbol{0}, \sigma^2 I_d)$, and the teacher weight matrix $W \in \mathbb{R}^{d \times d}$ is column-wise s-sparse:*

$$W_{j,k} = \begin{cases} \frac{1}{\sqrt{s}} & j \in S_k, \\ 0 & j \notin S_k, \end{cases} \quad (j = 1, 2, \ldots, d), \tag{11}$$

*with $S_k \subseteq \{1, 2, \ldots, d\}$ a random subset of size $s$ chosen uniformly at random from the $d$ indices.*

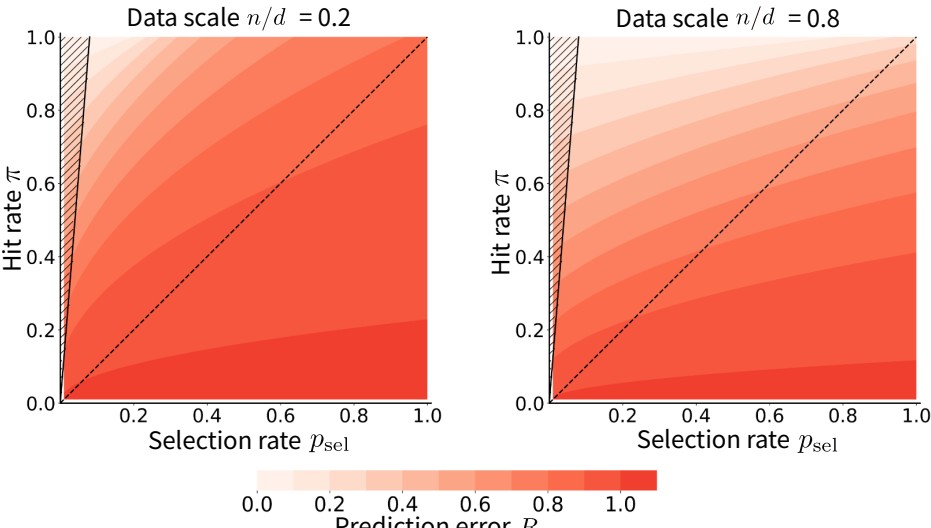

Figure 6: **Prediction error of sparse regression in the student–teacher model.** Each heatmap shows the prediction error $R$ of sparse regression with data scale $n/d = 0.2$ (left) and $n/d = 0.8$ (right). The horizontal axis represents selection rate $p_{\mathrm{sel}}$, and the vertical axis represents hit rate $\pi$. The diagonal line indicates the case where $p_{\mathrm{sel}} = \pi$, which corresponds to random selection. The upper-left shaded region defined by $\pi \geq p_{\mathrm{sel}}/a$ represents the $(p_{\mathrm{sel}}, \pi)$ values that are, in principle, unattainable when $a = 0.08$.

**Definition 1** (Generic sparse regression framework). *A generic sparse regression framework is defined as the following two-stage procedure. Given training data $\mathcal{D} = \{(\boldsymbol{x}^{(i)}, \boldsymbol{y}^{(i)})\}_{i=1}^{n}$ with $\boldsymbol{x}^{(i)} \in \mathbb{R}^{d_{\mathrm{in}}}$ and $\boldsymbol{y}^{(i)} \in \mathbb{R}^{d_{\mathrm{out}}}$, for each output coordinate $k \in \{1, \ldots, d_{\mathrm{out}}\}$:*

   *1. select a subset of inputs $\hat{S}_k \subseteq \{1, \ldots, d_{\mathrm{in}}\}$ with $|\hat{S}_k| = p_{\mathrm{sel}} d_{\mathrm{in}}$;*

   *2. fit ridge regression for $y_k$ using only $\boldsymbol{x}_{\hat{S}_k}$, with an optimally tuned penalty.*

Assumption 1 reflects the sparse structure of the brain-to-feature mapping, where each output dimension depends on only $s$ of the $d$ input dimensions (Fig. 5B). We denote the non-zero weight ratio by $a = s/d$, which represents the fraction of input dimensions that contribute to each output dimension; the smaller $a$ is, the sparser the input–output mapping. A framework in Definition 1 covers a broad class of sparse regression methods, including wrapper and filter methods. Fully embedded methods such as the Lasso (Tibshirani, 1996) or ARD (MacKay, 1992) do not fit this two-stage description exactly; however, refitting ridge regression on their selected supports aligns them with the above framework and is known to yield similar prediction performance (Belloni & Chernozhukov, 2013).

Under this student–teacher setting, let $\pi \in [0, 1]$ denote the hit rate of the variable-selection step, i.e., the fraction of the $s$ truly relevant inputs that are selected. Throughout the paper, the notation $\approx$ denotes asymptotic equivalence; the difference between the two sides converges to zero as $n, d \to \infty$ with $n/d \to \gamma$. By the asymptotic theory of Dicker (2016) for ridge regression, the following corollary holds.

**Corollary 1** (Asymptotic prediction error of post-selection ridge). *Under Assumption 1, consider the estimator defined in Definition 1, and assume that its variable-selection step achieves hit rate $\pi \in [0, 1]$. Then, as $n, d \to \infty$ with $n/d \to \gamma \in (0, \infty)$, the prediction error $R = \frac{1}{d}\mathbb{E}\left[\|\hat{\boldsymbol{y}}_{\mathrm{te}} - \boldsymbol{y}_{\mathrm{te}}\|^2\right]$ satisfies*

$$R \approx \sigma_{\mathrm{eff}}^2 \left\{ 1 + \frac{1}{2\rho}\left[ \tau^2(\rho - 1) - \rho + \sqrt{\left(\tau^2(\rho - 1) - \rho\right)^2 + 4\rho^2\tau^2} \right] \right\}, \tag{12}$$

*where*

$$\sigma_{\mathrm{eff}}^2 = 1 - \pi + \sigma^2, \qquad \tau^2 = \frac{\pi}{\sigma_{\mathrm{eff}}^2}, \qquad \rho = \frac{p_{\mathrm{sel}}}{\gamma}. \tag{13}$$

A proof is given in Appendix B.5. This expression allows us to estimate the prediction error $R$ from the selection rate $p_{\text{sel}}$ and the hit rate $\pi$ for a broad class of sparse regression methods. In the special case $p_{\text{sel}} = 1$ and $\pi = 1$ (all $d$ inputs selected), this expression reduces to the prediction error of ridge regression without variable selection. Fig. 6 shows heatmaps of the prediction error $R$ for sparse regression across combinations of the selection rate $p_{\text{sel}}$ and the hit rate $\pi$. With $p_{\text{sel}}$ fixed, $R$ decreases as $\pi$ increases. Compared with the no-selection baseline $(p_{\text{sel}}, \pi) = (1, 1)$, random selection along the diagonal $p_{\text{sel}} = \pi$ does not reduce the error, whereas variable selection that attains a high hit rate with a small $p_{\text{sel}}$ does reduce the error.

Sparse regression can exploit a sparse brain-to-feature mapping and is not structurally constrained by ODC. Our analysis within the student–teacher framework shows that variable selection, which achieves a high hit rate, can lead to accurate prediction even at small data scales. Corollary 1 applies to a broad class of sparse translators beyond the specific correlation-based filter we study next.

## 7 Quantifying hit rate and prediction error in practical sparse regression

We derived a prediction error expression for a broad class of sparse regression methods in terms of the selection rate $p_{\text{sel}}$ and the hit rate $\pi$. We next focus on a simple correlation-based filter as a concrete variable selection method and analyze how it realizes $p_{\text{sel}}$ and $\pi$ under the student–teacher model. We first provide a new theory which relates $p_{\text{sel}}$, $\pi$, the non-zero weight ratio $a$, and the data scale $\gamma = n/d$ (with $d := d_{\text{in}} = d_{\text{out}}$ as assumed in Section 6). We then evaluate the validity and practical range of our theoretical analysis by comparing the theoretical prediction errors with those from simulations and fMRI brain-to-image reconstruction data.

Among the various variable selection methods, we focus on a simple correlation-based filter (akin to SIS (Fan & Lv, 2008)) as a concrete procedure (Fig. 7A). This method is computationally trivial compared to wrapper or embedded approaches, making it well-suited for reconstruction pipelines that must efficiently handle large-scale data. For each output dimension, the filter selects the $d_{\text{sel}} = p_{\text{sel}}d$ input variables based on correlation scores (see Appendix B.6 for details). Applying this procedure separately across output dimensions yields a column-wise sparse support, which may differ across outputs.

Under the student–teacher framework, we obtain the following asymptotic relation for the hit rate of a simple correlation-based filter.

**Theorem 1** (Asymptotic hit rate of a correlation-based filter). *Under Assumption 1, consider a correlation-based filter with its selection rate $p_{\text{sel}} \in (0, 1)$. As $n, d \to \infty$ with $n/d \to \gamma \in (0, \infty)$, the selection rate $p_{\text{sel}}$, the hit rate $\pi$, the non-zero weight ratio $a$, and the data scale $\gamma = n/d$ satisfy*

$$\pi \approx \Phi\left(\Phi^{-1}\left(\frac{\pi_-}{2}\right) + \alpha\right) + \Phi\left(\Phi^{-1}\left(\frac{\pi_-}{2}\right) - \alpha\right), \tag{14}$$

*with*

$$\pi_- = \frac{p_{\text{sel}} - a\pi}{1 - a}, \qquad \alpha = \sqrt{\frac{\gamma}{a(1 + \sigma^2)}}, \tag{15}$$

*where $\Phi$ is the cumulative distribution function of the standard normal distribution.*

A proof is given in Appendix B.7. This relation defines a fixed-point equation for the hit rate $\pi$; we can estimate $\pi$ by numerically solving Eq. 14 for $\pi$ given the selection rate $p_{\text{sel}}$, the non-zero weight ratio $a$, and the data scale $\gamma = n/d$. Fig. 7B visualizes this relation for two data scales, $n/d = 0.2$ and $n/d = 0.8$. For a given selection rate $p_{\text{sel}}$, the hit rate $\pi$ increases as the non-zero weight ratio $a$ decreases, and a larger data scale $n/d$ leads to a larger hit rate $\pi$. In particular, when $a$ is small (i.e., the mapping is sparse), high hit rates $\pi$ are attainable even at small $p_{\text{sel}}$.

We next theoretically estimate the prediction error of two models: naive linear regression, which does not perform variable selection, and sparse linear regression, which performs variable selection using a correlation-based filter. Under the student–teacher model (Fig. 5B), we vary the non-zero weight ratio $a$ and the data scale $n/d$, and estimate the corresponding prediction error by using Corollary 1 and Theorem 1. We fix the

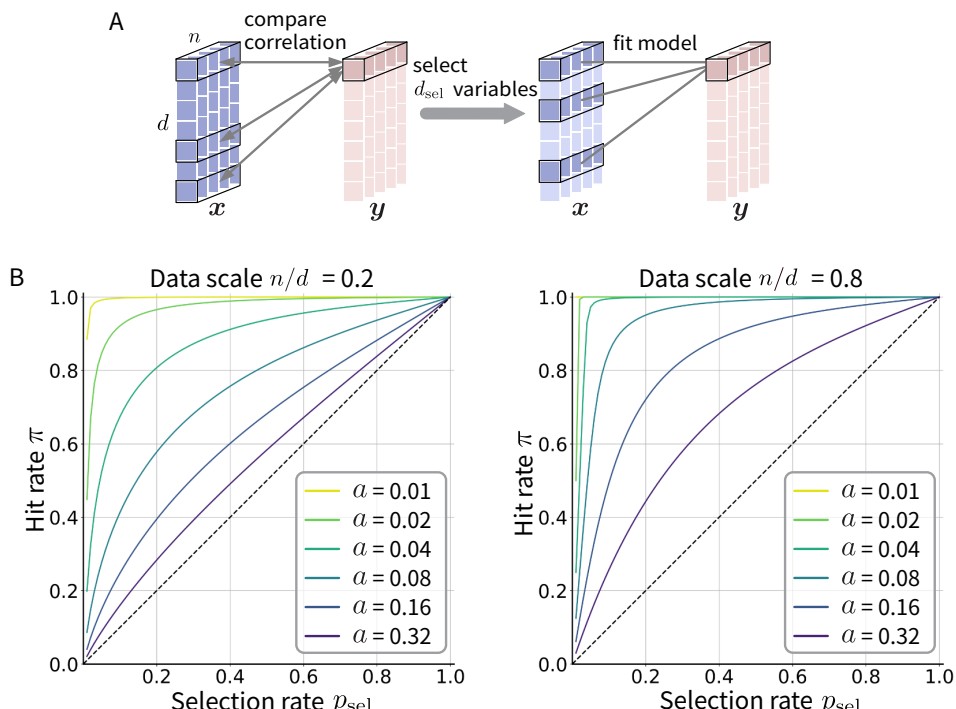

Figure 7: **Correlation-based variable selection and its hit rate under the student–teacher model.**
(A) Correlation-based filter for variable selection. The depth labeled $n$ is the sample axis: each slice represents one training sample. For each feature in $y$, compute its correlation with all inputs in $x$ across the $n$ samples, select the top $d_{\text{sel}} = p_{\text{sel}}d$ by absolute correlation, and fit the model using only the selected inputs. (B) Hit rate versus selection rate with a correlation-based filter. Each curve shows the relationship between the selection rate $p_{\text{sel}}$ and the hit rate $\pi$ for different non-zero weight ratios $a$ under the student–teacher model. The horizontal axis shows $p_{\text{sel}}$, and the vertical axis shows $\pi$. Left: data scale $n/d = 0.2$, right: data scale $n/d = 0.8$.

noise level at $\sigma = 0.1$ to facilitate comparison across conditions. For the sparse model, the selection rate $p_{\text{sel}}$ is set to minimize the estimated prediction error $R$ under Corollary 1 and Theorem 1. Fig. 8A summarizes the theoretical estimates. For naive linear regression, the prediction error decreases steadily with the data scale $n/d$ but remains insensitive to the sparsity $a$, so a single curve suffices; this confirms that the naive model cannot exploit the sparse structure of the data. For sparse linear regression, the prediction error decreases as the non-zero weight ratio $a$ becomes smaller. These results suggest that a sparse regression model can leverage the sparse structure of the data to achieve substantial accuracy gains and make accurate predictions even at small data scales when the mapping is sufficiently sparse.

To evaluate the finite-sample accuracy of our theory and its robustness beyond the modeling assumptions, we conduct simulations under four data generation processes (Fig. 8B). Specifically, we consider: (1) a baseline that exactly matches the assumptions used in our analysis (Fig. 5B); (2) Gaussian-distributed non-zero teacher weights that mimic uneven brain-to-feature mapping; (3) correlated signals generated with a Toeplitz covariance with parameter $\rho = 0.5$ to reflect the correlation structure in neural activity; and (4) input noise representing neural variability and measurement noise (see Appendix B.8 for details). Across the baseline, Gaussian weights, and input-noise settings, the theoretical estimates closely match the simulated prediction errors. For the correlated-signal setting, the theory and simulations align well within a practical range of data scales $n/d$, but we observe an increase in error at large data scales. This error increase is plausibly because a correlated signal makes variable selection more difficult, so the selection rate $p_{\text{sel}}$ that is optimal in the uncorrelated case fails to select the correct variables under correlation. These results show that our theoretical error expressions remain accurate in finite samples and are robust to deviations from

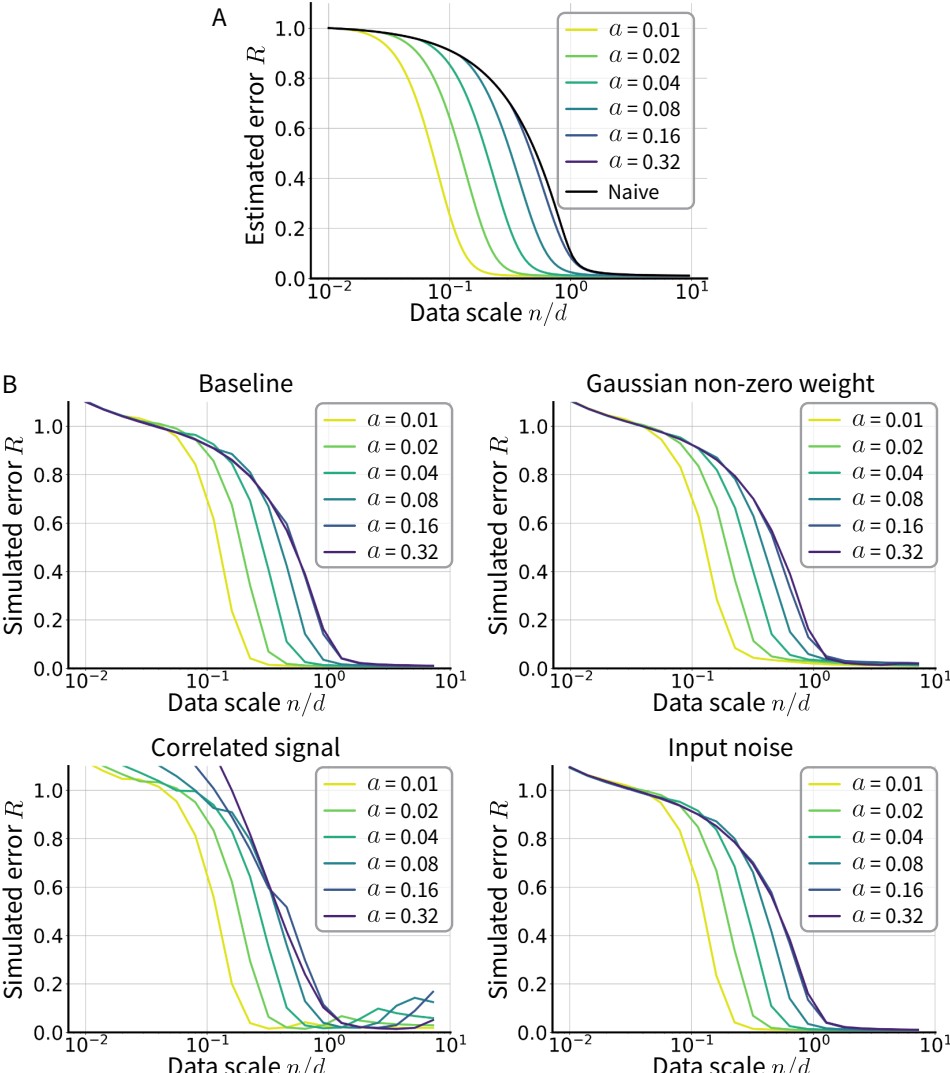

Figure 8: **The theoretical and simulated prediction error.** (A) The estimated prediction error in the student–teacher model. The black line shows the theoretical prediction error of naive linear regression and is independent of the non-zero weight ratio $a$, so only one curve is displayed. The colored lines show the theoretical prediction error of sparse regression for different non-zero weight ratios $a$. The horizontal axis represents the data scale $n/d$, and the vertical axis represents the estimated prediction error $R$. (B) The simulated prediction error under four data generation processes. Each plot shows the simulated prediction error $R$ of sparse regression under four different data generation processes: (1) Baseline (the same as the setting in our theory), (2) Gaussian-distributed non-zero weights, (3) Correlated signals modeled by a Toeplitz covariance with $\rho = 0.5$, (4) Input noise. The horizontal axis represents the data scale $n/d$, and the vertical axis represents the simulated prediction error $R$. Each colored line shows the prediction error for a different non-zero weight ratio $a$.

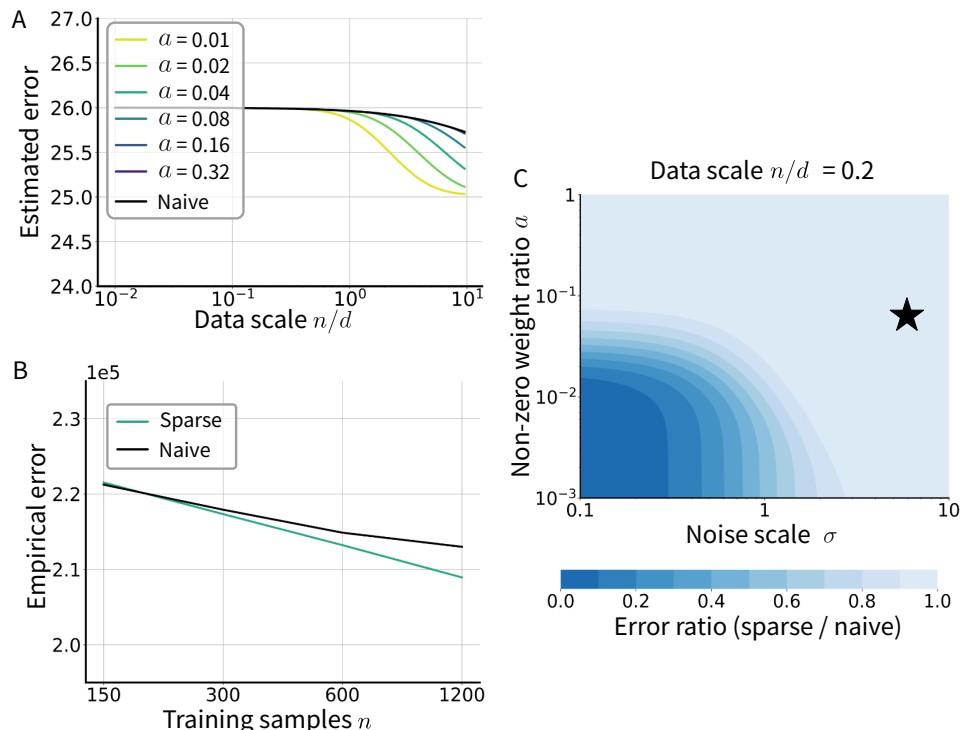

Figure 9: **Theoretical and empirical prediction errors with a sensitivity map of the sparse advantage.** (A) Theoretical prediction error of naive and sparse regression as a function of the data scale $\gamma = n/d$ under the student–teacher model. Each curve corresponds to a different non-zero weight ratio $a$, and the noise level is set to $\sigma = 5$. (B) Empirical prediction errors of naive and sparse translators on the Deeprecon dataset (Shen et al., 2019) with VGG19 target features as a function of the number of training samples $n \in \{150, 300, 600, 1200\}$. Since $d_{\text{out}}$ is very large, these $n$ values correspond to small data scales $\gamma = n/d_{\text{out}} < 1$. Panels A and B use different error scales, so their absolute magnitudes are not directly comparable; we instead compare the relative errors of naive and sparse regression. The empirical errors show only a modest advantage of sparse regression over naive regression. (C) Sensitivity map showing the relative advantage of sparse regression at a representative small data scale ($\gamma = 0.2$): the heatmap reports the predicted error ratio $R_{\text{sparse}}/R_{\text{naive}}$ as a function of the noise level $\sigma$ and the non-zero weight ratio $a$. Values close to 1 indicate a modest benefit of sparsity. The star marks a reference regime motivated by Deeprecon ($\sigma \approx 5$; $a$ unknown).

the modeling assumptions that reflect properties of real data. They indicate that our framework provides reliable and informative guidance in practical settings.

Finally, we provide an exploratory link between our theory and the empirical regime of real fMRI decoding on the Deeprecon dataset (Shen et al., 2019). We vary the number of training samples $n$ over the range $\{150, 300, 600, 1200\}$ and evaluate the prediction error of both naive and sparse linear regression. The dimensionality of the latent features is on the order of $d_{\text{out}} \approx 10^6$, so these training sample sizes correspond to small data scales $\gamma = n/d_{\text{out}} < 1$, even when accounting for redundancy in the latent representation. Existing fMRI datasets likely fall in a relatively high-noise regime. Since the trial-to-trial variability for repeated presentations is on the order of five in our noise-to-signal metric (see Appendix B.9), we use $\sigma = 5$ as a convenient reference value to illustrate this regime and interpret the real-data comparison qualitatively. The true non-zero weight ratio $a$ is unknown, so we evaluate the theoretical error over a range of plausible values of $a$. Fig. 9A shows the theoretical errors of naive and sparse regression, and Fig. 9B shows the empirical prediction errors on the Deeprecon dataset with VGG19 target features (see Appendix B.10 for details). Panels A and B use different error scales, so their absolute magnitudes are not directly comparable; we instead compare the relative errors of naive and sparse regression. Within the range of $\gamma$, $\sigma$, and $a$

compatible with this dataset, our analysis predicts that sparse regression yields only a modest reduction in prediction error, while a substantial part of the error remains irreducible due to noise. Consistent with this prediction, the empirical results show only a modest error reduction of sparse regression relative to naive regression. As a reference, sparse regression does not outperform naive regression with CLIP target features, which are often regarded as a distributed representation (Fig. A7). Fig. 9C shows a sensitivity map of the predicted relative advantage of sparse regression as the error ratio $R_{\mathrm{sparse}}/R_{\mathrm{naive}}$ over the noise level $\sigma$ and the non-zero weight ratio $a$ at a representative small data scale. The star marks a reference regime motivated by Deeprecon ($\sigma \approx 5$; $a$ unknown), which lies in a region where the predicted ratio is close to 1. This map is consistent with the modest empirical gain on current datasets and suggests that further reductions in prediction error will require lowering measurement noise beyond what can be achieved by sparsity alone.

## 8 Discussion

In this study, we analyzed the behavior of translators in brain-to-image reconstruction pipelines at small data scales. We first clarified how small data scales induce output dimension collapse in naive multivariate linear regression, and we evaluated its impact on the reconstruction pipelines that employ a naive translator. We then analyzed sparse linear regression in a student–teacher model and derived an expression for the prediction error in terms of data scale, sparsity, and related parameters. Our results suggest that a sparse model can achieve accurate prediction even at small data scales by exploiting the sparse brain-to-feature structure. These findings imply that sparse regression does not suffer from ODC; its gains do not arise from improving prediction within the training subspace but from enabling predictions that extend beyond that subspace.

In Section 4, we derived the best prediction for naive multivariate linear regression in the translator. This best prediction gives the lowest possible latent-feature error, so no naive translator can outperform it even under noise-free brain activity. As a caveat, the smallest latent-feature error need not correspond to the most accurate reconstructed image after applying the generator because latent-feature error and image-space error can diverge. Even so, the analysis makes clear that collapse at small data scales imposes a substantial irreducible error on the latent features, which directly limits the reconstruction quality. In addition, the best prediction analysis is valuable not only for characterizing ODC on small data scales but also for diagnosing ODC by the dataset bias described by Shirakawa et al. (2025). The calculation depends only on the training and test images (no brain data is required), so it can be performed prior to any neuroimaging experiment. Researchers can confirm and, if necessary, adjust their stimulus sets before collecting costly neural recordings.

In Section 6 and Section 7, we analyzed sparse linear regression in a student–teacher model and compared its predictions with simulations and the Deeprecon dataset. The theory predicts substantial accuracy gains when the noise level is moderate and the brain-to-feature mapping is sufficiently sparse. Current brain-to-image reconstruction datasets depart from this regime: the estimated noise level is high, and the sparsity of the true mapping is likely limited, which can explain why the empirical gains on real data are modest. In particular, strong sparsity can be expected only for the "brain-like" latent features that reflect locality or selectivity in visual cortical representations, such as localized bases or intermediate convolutional feature maps. In contrast, the latent features used in current reconstruction studies, including CLIP embeddings and fully connected CNN layers, often encode information in a distributed manner (i.e., a feature is represented by a pattern across many units rather than a single unit), which may not fully satisfy the sparsity assumption. Our results do not suggest that sparsity always improves performance, but provide a quantitative diagnostic of when it can and cannot help. These results do not provide an immediate recipe for large performance improvements on existing datasets, but instead offer a quantitative guideline for future choices of measurement procedures and feature representation design: further progress will require both improved denoising of neural recordings and the construction of AI feature spaces that more closely approximate the brain's representations, so that strong sparsity can be realized and effectively exploited.

Although we examined only linear translators, several recent reconstruction methods have adopted nonlinear architectures as the translator module (Chen et al., 2023; Scotti et al., 2023; 2024). In principle, nonlinear networks can encode additional inductive biases, which may be attractive in data-scarce settings. However, simply increasing model capacity does not eliminate ODC as a concern. In the neural-tangent-kernel (NTK)

regime (Jacot et al., 2018), nonlinear models behave like kernel linear models, so they cannot avoid ODC. Empirically, Shirakawa et al. (2025) reported that nonlinear translators used for reconstruction also experience collapse-like phenomena. A rigorous theoretical analysis of whether nonlinear translators can genuinely support zero-shot prediction remains important for future work.

Furthermore, while our study focused on the translator, the generator module might partially compensate for inaccuracies in latent feature prediction. In the latest method of Shen et al. (2019), the improved visual appearance arises not only from replacing naive linear regression with sparse regression but also from generator-side refinements, such as stronger image priors, latent feature scaling, and carefully chosen feature inversion losses. More generally, it has been observed that a generator can recover a visually similar image even under substantial perturbations of the latent features (Onoo et al., 2026; Lee et al., 2025). This observation suggests that high image-space accuracy may be achievable even when latent-feature error remains significant. More recently, some reconstruction pipelines have begun to adopt diffusion models as the generator (Ozcelik & VanRullen, 2023; Cheng et al., 2023; Scotti et al., 2024). These generators can produce photorealistic images even from inaccurate latent features. However, this approach carries the risk of introducing information not actually present in the brain's representation. As illustrated in Fig. A6, a visually plausible reconstruction may nonetheless depart from the perceptual content encoded in neural activity. The "Recovery Check", which verifies reconstructions obtained from the ground-truth features, can ensure that only the information contained in the latent features is reconstructed (Shirakawa et al., 2025).

Finally, our results apply to data-limited prediction problems at the interface of neuroscience and machine learning, including brain decoding and brain encoding across vision, speech, and language. For example, Brain-Score assesses brain-DNN alignment by fitting linear encoding models from model features to neural responses (Schrimpf et al., 2020). When solving regression with small datasets, ODC is a critical concern. Sparse regression can mitigate ODC; however, in encoding settings, the selected DNN features tend to exhibit substantial overlap across voxels (Nonaka et al., 2021). Although we do not treat this situation here, understanding how such selection concentration shapes prediction remains an important direction for future work.

The significance of our study lies in its dual contribution. First, we showed that ODC can be a practical bottleneck for naive linear translators in reconstruction pipelines at small data scales, restricting generalization beyond the training data. Second, within a student–teacher framework, we derived a prediction error expression for sparse regression and clarified when accurate prediction is achievable at small data scales. These contributions provide a method for diagnosing ODC in existing reconstruction pipelines and offer quantitative guidelines for advancing brain-to-image reconstruction without relying on large-scale data.

### Broader Impact Statement

This work provides a theoretical and diagnostic analysis that can improve the scientific validity of brain-to-image decoding under limited data. At the same time, because brain decoding could ultimately connect to the externalization of subjective experience (i.e., "mind reading"), uses beyond research settings raise important privacy considerations. Accordingly, any future applications should be considered only under robust governance and with fully informed consent from participants. Moreover, reconstruction outcomes depend on the stimulus set, modeling assumptions, measurement noise, and the generative prior of the image generator, so claims about recovered perceptual content should be made conservatively.

### Code availability

The code used to reproduce the experiments in this paper is publicly available at `https://github.com/KamitaniLab/OvercomingOutputDimensionCollapse`.

### Acknowledgments

We thank our laboratory team, especially Ken Shirakawa and Hideki Izumi, for their invaluable feedback and insightful suggestions on the manuscript. This work was supported by JSPS KAKENHI (JP20H05705 and JP25K24743 to Y.K.; JP21K17821 to Y.N.), JST CREST (JPMJCR22P3 to Y.K.), JST BOOST (JP-

MJBS2407 to K.O.), AMED (JP24wm0625409 to Y.K.), and the Guardian Robot Project, RIKEN (to Y.K.).

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

# A   Supplementary Figures

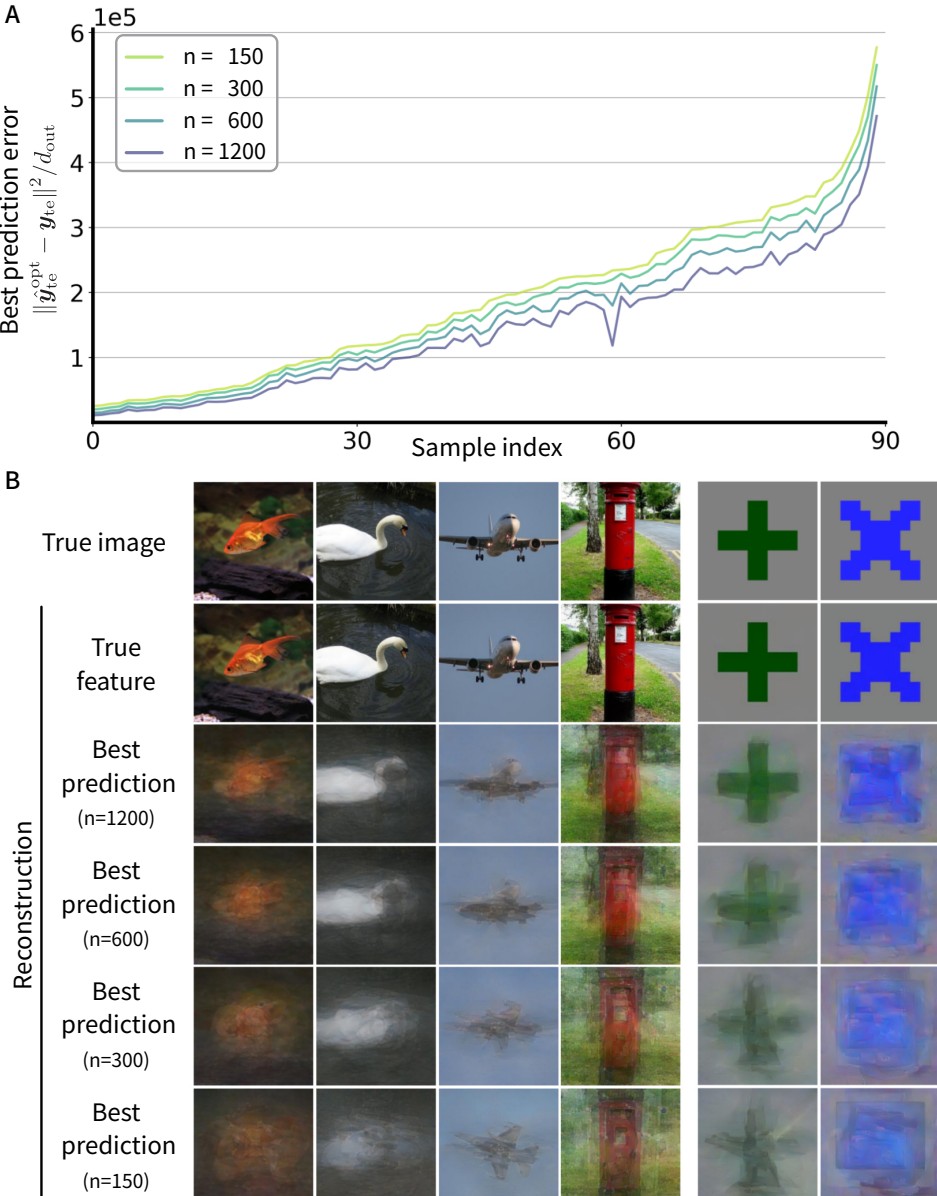

Figure A1: **The best predictions for different training data sizes in a category-overlap setting.** To disentangle data-scale effects from dataset bias, we evaluate a setting where the training and test sets share image categories. (A) The best prediction error $\frac{1}{d_{\text{out}}}\|\hat{\boldsymbol{y}}_{\text{te}}^{\text{opt}} - \boldsymbol{y}_{\text{te}}\|^2$ for each test sample across training data sizes 1200, 600, 300, and 150. The vertical axis represents the error of the best prediction, and the horizontal axis represents the sample index sorted by the error at $n = 150$. (B) Representative reconstructions from the best predictions for different training data sizes. From top: ground truth, reconstructed images generated from the true features $\mathcal{G}(\boldsymbol{y}_{\text{te}})$, the best predictions $\mathcal{G}(\hat{\boldsymbol{y}}_{\text{te}}^{\text{opt}})$ using 1200, 600, 300, and 150 samples. The left four columns show natural images, and the right two columns show artificial shape images. A similar result in the category-overlap setting confirms that limited data scale alone can induce ODC, even when dataset-bias effects are minimized.

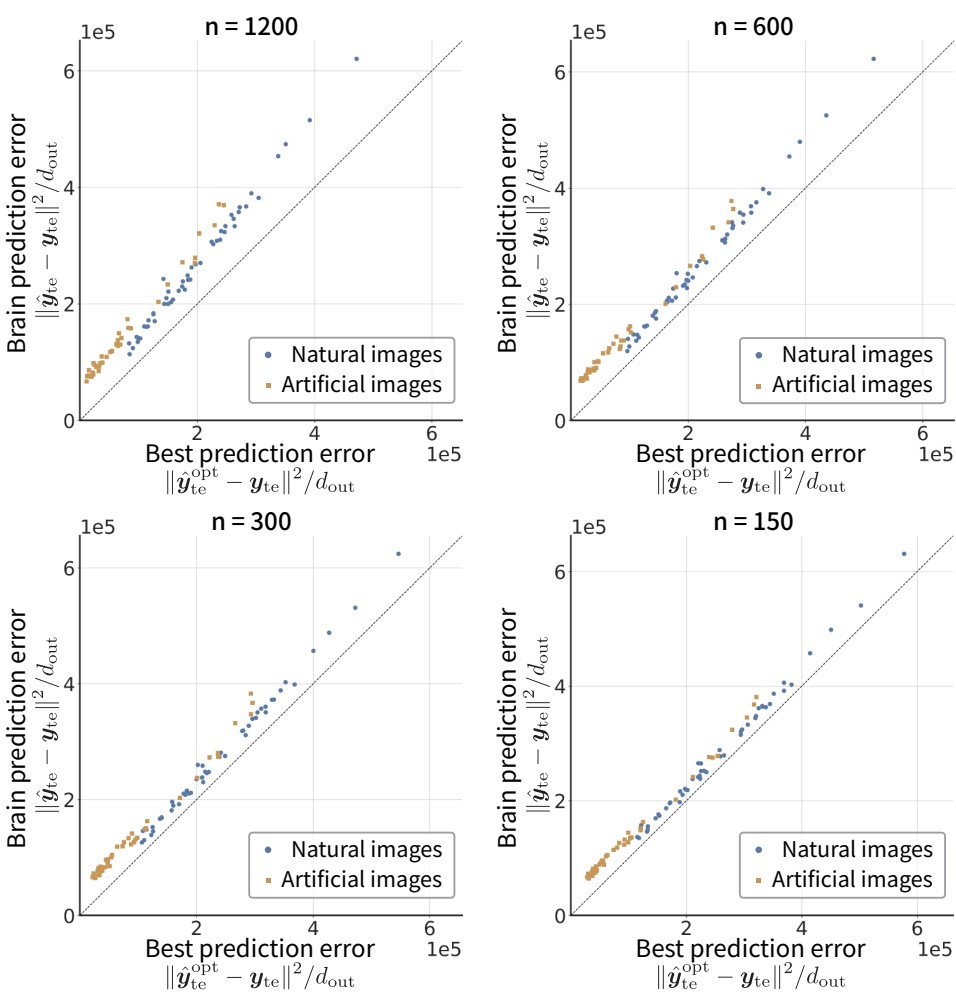

Figure A2: **Comparison of the best prediction and the brain prediction for subsampled training sets.** Each panel shows best prediction error $\frac{1}{d_{\text{out}}}\|\hat{\boldsymbol{y}}_{\text{te}}^{\text{opt}} - \boldsymbol{y}_{\text{te}}\|^2$ and brain prediction error $\frac{1}{d_{\text{out}}}\|\hat{\boldsymbol{y}}_{\text{te}} - \boldsymbol{y}_{\text{te}}\|^2$ with subsampled training sets of 1200, 600, 300, and 150 samples. As the data scale decreases, the output dimension collapse intensifies, and the brain's prediction approaches the best prediction.

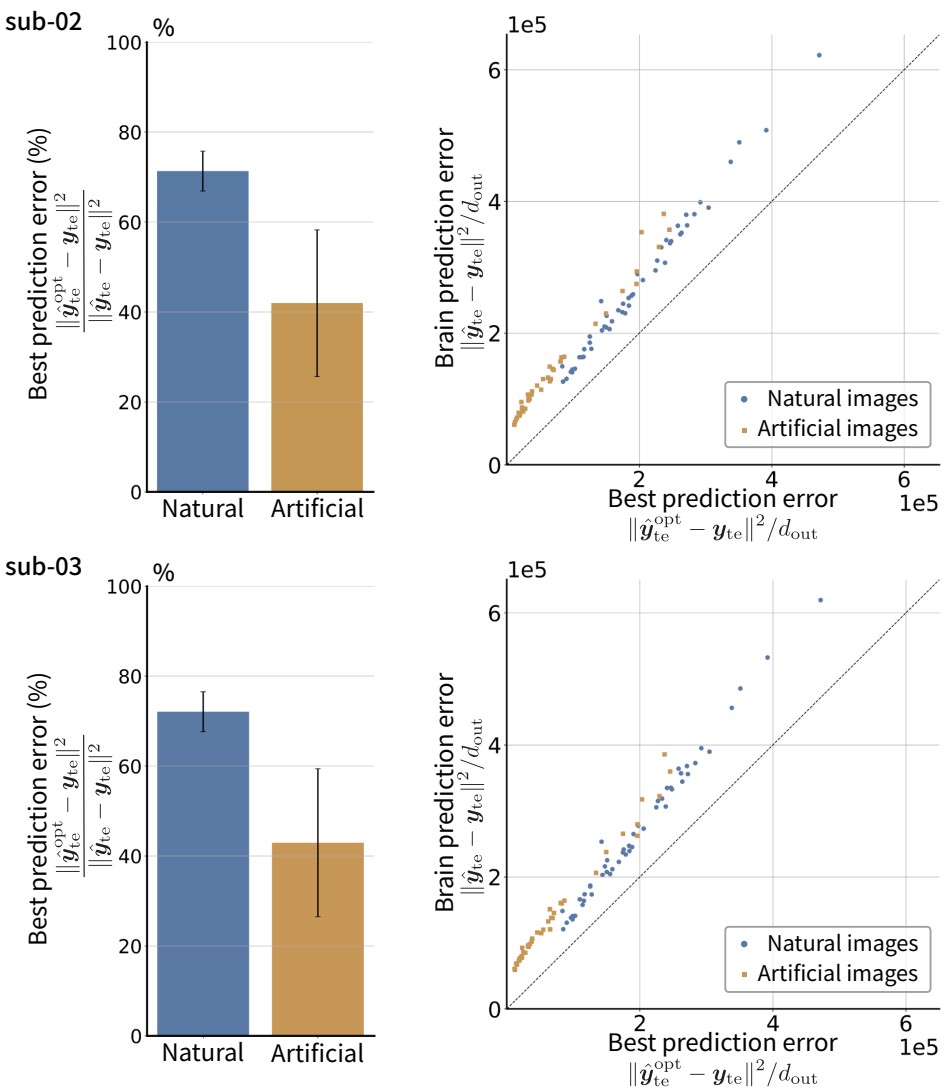

Figure A3: **Comparison of the best prediction and the brain prediction for additional subjects.** (A) Same as Fig. 4A (`sub-01`), shown for `sub-02` (top) and `sub-03` (bottom). Left: Percentage of the best prediction error relative to the brain prediction error $\|\hat{\boldsymbol{y}}_{\text{te}}^{\text{opt}} - \boldsymbol{y}_{\text{te}}\|^2 / \|\hat{\boldsymbol{y}}_{\text{te}} - \boldsymbol{y}_{\text{te}}\|^2$ (blue for natural images, orange for artificial shapes; error bars denote the standard deviation across samples). Right: Sample-wise comparison of best prediction error $\frac{1}{d_{\text{out}}}\|\hat{\boldsymbol{y}}_{\text{te}}^{\text{opt}} - \boldsymbol{y}_{\text{te}}\|^2$ and brain prediction error $\frac{1}{d_{\text{out}}}\|\hat{\boldsymbol{y}}_{\text{te}} - \boldsymbol{y}_{\text{te}}\|^2$ (each dot represents one image; the dotted line indicates equality). Similar results as in Fig. 4 are observed across subjects.

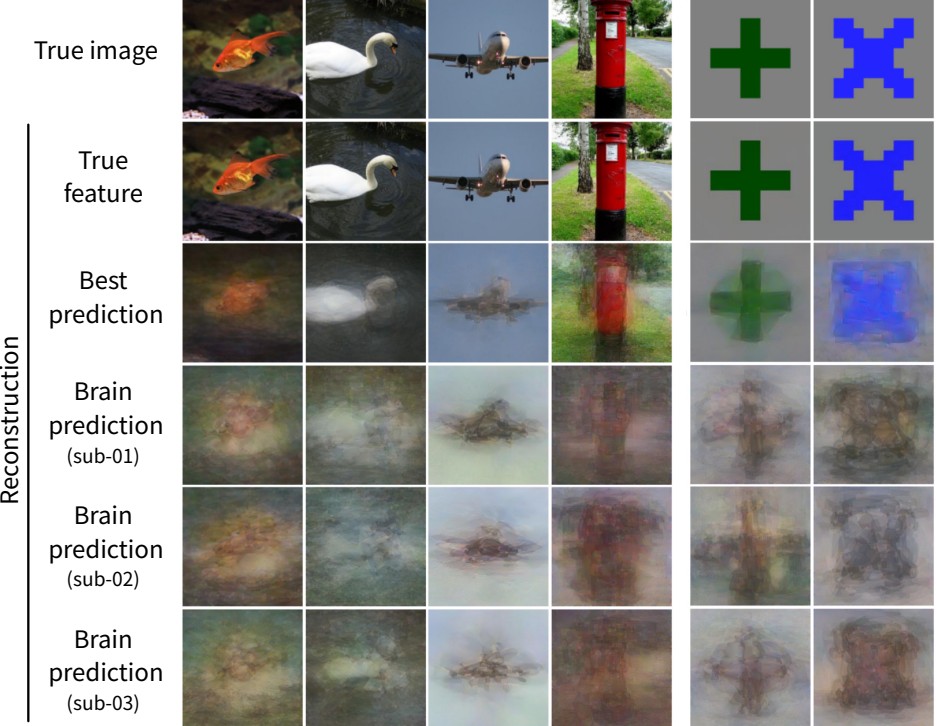

Figure A4: **Reconstructions from the best predictions and brain predictions across subjects.** Same as Fig. 4B (`sub-01`), shown for `sub-01`, `sub-02`, and `sub-03`. From top: ground truth, reconstructions from the true features $\mathcal{G}(\boldsymbol{y}_{\text{te}})$, reconstructions from the best prediction features $\mathcal{G}(\hat{\boldsymbol{y}}_{\text{te}}^{\text{opt}})$, and reconstructions from the brain prediction features $\mathcal{G}(\hat{\boldsymbol{y}}_{\text{te}})$ for `sub-01`, `sub-02`, and `sub-03`. Similar results as in Fig. 4 are observed across subjects.

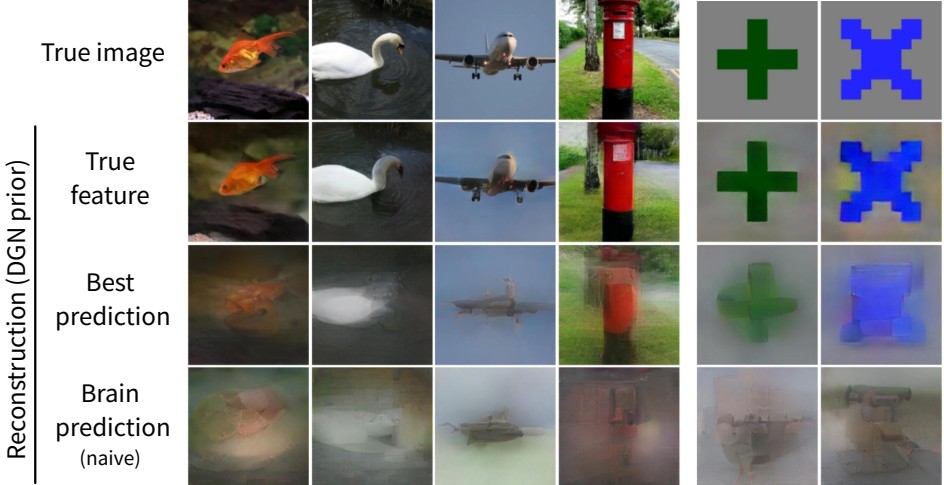

Figure A5: **Reconstructions from the best predictions and brain predictions with a DGN image prior.** Same as Fig. 4B, except that the generator's image prior is replaced with Deep Generator Network (DGN) (Dosovitskiy & Brox, 2016). From top: ground truth, reconstructions from the true features $\mathcal{G}(\boldsymbol{y}_{\text{te}})$, the best predictions $\mathcal{G}(\hat{\boldsymbol{y}}_{\text{te}}^{\text{opt}})$, and brain predictions $\mathcal{G}(\hat{\boldsymbol{y}}_{\text{te}})$. The left four columns show natural images, and the right two columns show artificial shape images. Similar results as in Fig. 4B are observed even with a stronger image prior.

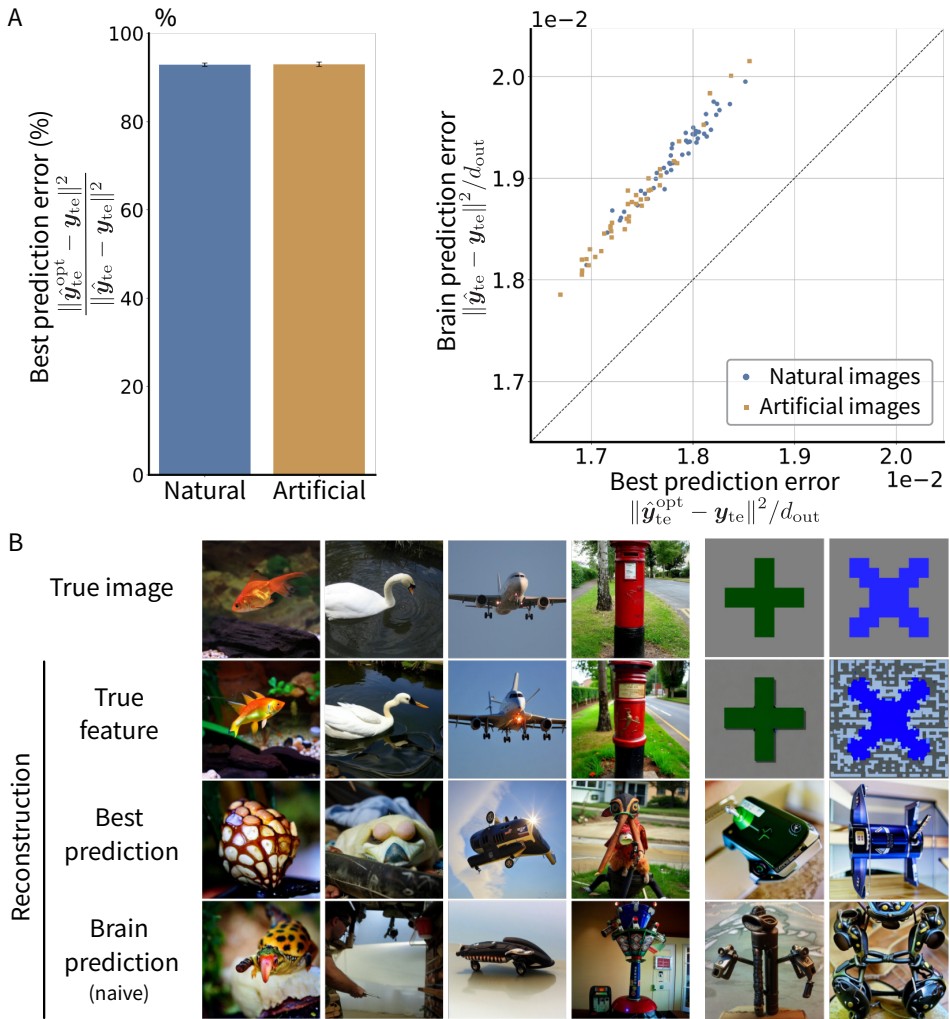

Figure A6: **Comparison of the best prediction and the brain prediction on another reconstruction method (Ozcelik & VanRullen, 2023).** (A) Left: Percentage of the best prediction error to the brain prediction error $\|\hat{\boldsymbol{y}}_{\text{te}}^{\text{opt}} - \boldsymbol{y}_{\text{te}}\|^2 / \|\hat{\boldsymbol{y}}_{\text{te}} - \boldsymbol{y}_{\text{te}}\|^2$. Blue for natural images, orange for artificial shapes. The error bars denote the standard deviation across samples. Right: Sample-wise comparison of best prediction error $\frac{1}{d_{\text{out}}}\|\hat{\boldsymbol{y}}_{\text{te}}^{\text{opt}} - \boldsymbol{y}_{\text{te}}\|^2$ and brain prediction error $\frac{1}{d_{\text{out}}}\|\hat{\boldsymbol{y}}_{\text{te}} - \boldsymbol{y}_{\text{te}}\|^2$. Each dot represents one image, and the dotted line indicates where the two errors are equal. With this CLIP-and-VDVAE-based latent representation, the irreducible (out-of-subspace) error accounts for an even larger fraction of the brain prediction error. (B) Representative reconstructions generated from the best predictions and brain predictions. From top: ground truth, reconstructions from the true features $\mathcal{G}(\boldsymbol{y}_{\text{te}})$, the best predictions $\mathcal{G}(\hat{\boldsymbol{y}}_{\text{te}}^{\text{opt}})$, brain predictions $\mathcal{G}(\hat{\boldsymbol{y}}_{\text{te}})$. Even with a strong diffusion-based generator, reconstructions from both predictions are photorealistic but still differ from the ground truth.

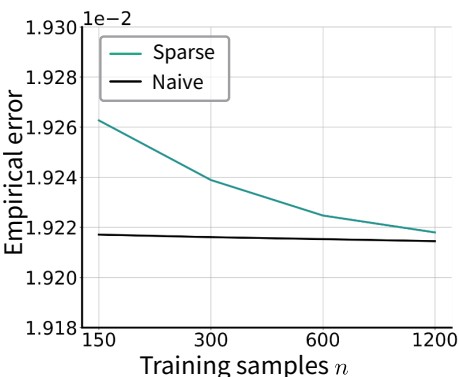

Figure A7: **Empirical prediction errors on Deeprecon with CLIP target features.** Empirical prediction errors of naive and sparse translators on the Deeprecon dataset (Shen et al., 2019) as a function of the number of training samples $n \in \{150, 300, 600, 1200\}$, using CLIP target features in place of the VGG19 target features used in Fig. 9(B). The naive and sparse errors are nearly identical; although the sparse translator may appear substantially worse in the plot, this apparent gap is visually exaggerated by the narrow y-axis range. With CLIP target features, which are often regarded as a distributed representation, sparse regression does not achieve lower error than naive regression.

# B   Supplementary Materials

## B.1   Output subspace restriction in kernel linear regression and PLS

We show that kernel linear regression and PLS produce predictions that are linear combinations of the training outputs $\{\boldsymbol{y}^{(i)}\}_{i=1}^{n}$. Let $(\boldsymbol{x}^{(i)}, \boldsymbol{y}^{(i)})$ for $i = 1, \ldots, n$ be training pairs with $\boldsymbol{x}^{(i)} \in \mathbb{R}^{d_{\mathrm{in}}}$ and $\boldsymbol{y}^{(i)} \in \mathbb{R}^{d_{\mathrm{out}}}$. Let $Y_{\mathrm{tr}} \in \mathbb{R}^{n \times d_{\mathrm{out}}}$ be the training output matrix and define $\mathcal{A} := \{Y_{\mathrm{tr}}^{\top} \boldsymbol{m} \mid \boldsymbol{m} \in \mathbb{R}^n\}$.

**Kernel linear regression.**   Let $k(\cdot, \cdot)$ be a positive semidefinite kernel and $K \in \mathbb{R}^{n \times n}$ the Gram matrix with entries $K_{ij} = k(\boldsymbol{x}^{(i)}, \boldsymbol{x}^{(j)})$. For a test input $\boldsymbol{x}_{\mathrm{te}}$, define $\boldsymbol{k}_{\mathrm{te}} = [k(\boldsymbol{x}_{\mathrm{te}}, \boldsymbol{x}^{(1)}), \ldots, k(\boldsymbol{x}_{\mathrm{te}}, \boldsymbol{x}^{(n)})]^{\top}$. Kernel linear regression fits functions in the RKHS and, by the representer theorem, the multi-output predictor takes the matrix form:

$$\hat{\boldsymbol{y}}_{\mathrm{te}} = A^{\top} \boldsymbol{k}_{\mathrm{te}} \quad \text{with} \quad A = (K + \lambda I)^{-1} Y_{\mathrm{tr}}.$$

Therefore,

$$\hat{\boldsymbol{y}}_{\mathrm{te}} = Y_{\mathrm{tr}}^{\top} (K + \lambda I)^{-1} \boldsymbol{k}_{\mathrm{te}} = Y_{\mathrm{tr}}^{\top} \boldsymbol{m} = \sum_{i=1}^{n} m_i \, \boldsymbol{y}^{(i)},$$

where $\boldsymbol{m} = (K + \lambda I)^{-1} \boldsymbol{k}_{\mathrm{te}} \in \mathbb{R}^n$. Thus, $\hat{\boldsymbol{y}}_{\mathrm{te}} \in \mathcal{A}$, i.e., every prediction is a linear combination of the training outputs, independent of the particular choice of kernel.

**Partial least squares (PLS).**   We consider multivariate PLS with $r$ components on $(X_{\mathrm{tr}}, Y_{\mathrm{tr}})$. Two standard variants are NIPALS and SIMPLS, which differ in how they construct the training score matrix $T \in \mathbb{R}^{n \times r}$ and the out-of-sample score map $t(\boldsymbol{x}_{\mathrm{te}}) \in \mathbb{R}^r$, yet the prediction formula below does not depend on which algorithm produced $T$ and $t(\cdot)$.

Given $T$, the Y-side loading follows from least squares:

$$Q^{\top} = \underset{B \in \mathbb{R}^{r \times d_{\mathrm{out}}}}{\operatorname{argmin}} \| Y_{\mathrm{tr}} - TB \|_F^2 = (T^{\top} T)^{-1} T^{\top} Y_{\mathrm{tr}}.$$

For a test input $\boldsymbol{x}_{\mathrm{te}}$, PLS predicts

$$\hat{\boldsymbol{y}}_{\mathrm{te}} = t(\boldsymbol{x}_{\mathrm{te}})^{\top} Q^{\top} = t(\boldsymbol{x}_{\mathrm{te}})^{\top} (T^{\top} T)^{-1} T^{\top} Y_{\mathrm{tr}} = Y_{\mathrm{tr}}^{\top} \boldsymbol{m},$$

where $\boldsymbol{m} = T(T^{\top} T)^{-1} t(\boldsymbol{x}_{\mathrm{te}}) \in \mathbb{R}^n$. Thus, $\hat{\boldsymbol{y}}_{\mathrm{te}} \in \mathcal{A}$. This conclusion holds for NIPALS and SIMPLS alike, regardless of the specific construction of $T$ and $t(\cdot)$.

## B.2   Derivation of the best prediction for naive linear regression

Let $\boldsymbol{y}_{\mathrm{te}} \in \mathbb{R}^{d_{\mathrm{out}}}$ be a test output and $Y_{\mathrm{tr}} \in \mathbb{R}^{n \times d_{\mathrm{out}}}$ be the training output matrix. The best prediction in naive multivariate linear regression is derived analytically as follows:

$$\hat{\boldsymbol{y}}_{\mathrm{te}}^{\mathrm{opt}} = \underset{\hat{\boldsymbol{y}}_{\mathrm{te}} \in \mathcal{A}}{\operatorname{argmin}} \|\hat{\boldsymbol{y}}_{\mathrm{te}} - \boldsymbol{y}_{\mathrm{te}}\|^2$$
$$= Y_{\mathrm{tr}}^{\top} \left(Y_{\mathrm{tr}} Y_{\mathrm{tr}}^{\top}\right)^{\dagger} Y_{\mathrm{tr}} \boldsymbol{y}_{\mathrm{te}},$$

where $A^{\dagger}$ is the Moore-Penrose pseudo-inverse of $A$.

*Proof.* Since $\mathcal{A} = \left\{Y_{\mathrm{tr}}^{\top} \boldsymbol{m} \mid \boldsymbol{m} \in \mathbb{R}^n\right\}$, we can express the best prediction $\hat{\boldsymbol{y}}_{\mathrm{te}}^{\mathrm{opt}}$ as

$$\hat{\boldsymbol{y}}_{\mathrm{te}}^{\mathrm{opt}} = \underset{\hat{\boldsymbol{y}}_{\mathrm{te}} \in \mathcal{A}}{\operatorname{argmin}} \|\hat{\boldsymbol{y}}_{\mathrm{te}} - \boldsymbol{y}_{\mathrm{te}}\|^2$$
$$= \underset{\boldsymbol{m} \in \mathbb{R}^n}{\operatorname{argmin}} \|Y_{\mathrm{tr}}^{\top} \boldsymbol{m} - \boldsymbol{y}_{\mathrm{te}}\|^2$$

The normal equation obtains the solution:

$$Y_{\mathrm{tr}} Y_{\mathrm{tr}}^\top \boldsymbol{m} = Y_{\mathrm{tr}} \boldsymbol{y}_{\mathrm{te}}.$$

Thus, the best prediction is

$$\hat{\boldsymbol{y}}_{\mathrm{te}}^{\mathrm{opt}} = Y_{\mathrm{tr}}^\top \left( Y_{\mathrm{tr}} Y_{\mathrm{tr}}^\top \right)^\dagger Y_{\mathrm{tr}} \boldsymbol{y}_{\mathrm{te}}.$$

$\square$

### B.3 Practical recipe for computing the best prediction

To compute the best prediction $\hat{\boldsymbol{y}}_{\mathrm{te}}^{\mathrm{opt}}$ for a test stimulus, we use only the training-output matrix $Y_{\mathrm{tr}} \in \mathbb{R}^{n \times d_{\mathrm{out}}}$ and the test latent feature vector $\boldsymbol{y}_{\mathrm{te}} \in \mathbb{R}^{d_{\mathrm{out}}}$. In particular, this computation does not require the training inputs $X_{\mathrm{tr}}$ nor the test brain activity $\boldsymbol{x}_{\mathrm{te}}$. We provide pseudocode below to compute the best prediction.

---

**Algorithm 1** Best prediction for multiple test stimuli

---

**Input:** $Y_{\mathrm{tr}} \in \mathbb{R}^{n \times d_{\mathrm{out}}}$, test features $\{\boldsymbol{y}_{\mathrm{te}}^{(j)}\}_{j=1}^{n_{\mathrm{te}}}$, scaling parameters $\theta$

**Output:** best predictions $\{\hat{\boldsymbol{y}}_{\mathrm{te}}^{\mathrm{opt},(j)}\}_{j=1}^{n_{\mathrm{te}}}$

1: $\tilde{Y}_{\mathrm{tr}} \leftarrow \mathrm{scale}(Y_{\mathrm{tr}}; \theta)$
2: $G \leftarrow \tilde{Y}_{\mathrm{tr}} \tilde{Y}_{\mathrm{tr}}^\top$
3: $G^\dagger \leftarrow \mathrm{pinv}(G)$
4: **for** $j = 1$ **to** $n_{\mathrm{te}}$ **do**
5: $\quad \tilde{\boldsymbol{y}}_{\mathrm{te}}^{(j)} \leftarrow \mathrm{scale}(\boldsymbol{y}_{\mathrm{te}}^{(j)}; \theta)$
6: $\quad \hat{\boldsymbol{y}}_{\mathrm{te}}^{\mathrm{opt},(j)} \leftarrow \tilde{Y}_{\mathrm{tr}}^\top \Big( G^\dagger \big( \tilde{Y}_{\mathrm{tr}} \tilde{\boldsymbol{y}}_{\mathrm{te}}^{(j)} \big) \Big)$
7: **end for**
8: **return** $\{\hat{\boldsymbol{y}}_{\mathrm{te}}^{\mathrm{opt},(j)}\}_{j=1}^{n_{\mathrm{te}}}$

---

We compute the best prediction via the $n \times n$ Gram matrix $G = Y_{\mathrm{tr}} Y_{\mathrm{tr}}^\top$. Because $G$ is an $n \times n$ matrix, $G^\dagger$ can be computed stably using standard numerical solvers even when $d_{\mathrm{out}}$ is extremely large. Note that if the outputs are centered by the scaler, then $\mathrm{rank}(G) \leq n - 1$. In Step 6, the update should be computed in a right-to-left order; otherwise one would have to materialize a $d_{\mathrm{out}} \times d_{\mathrm{out}}$ matrix, which is often too large to fit in memory.

The computation up to $G^\dagger$ costs $O(n^2 d_{\mathrm{out}} + n^3)$ time. When $d_{\mathrm{out}} \gg n$, the dominant cost is Step 2 (forming $G = Y_{\mathrm{tr}} Y_{\mathrm{tr}}^\top$). $G^\dagger$ can be reused for each new test stimulus, and the cost is $O(n d_{\mathrm{out}} + n^2)$ time.

### B.4 ODC analysis in the Deeprecon dataset

We analyzed output dimension collapse (ODC) on real fMRI data using the Deeprecon dataset (Shen et al., 2019). Unless stated otherwise, we used subject `sub-01` and the visual cortex (VC) ROI, and followed the preprocessing, latent feature extraction, and reconstruction protocol of Shen et al. (2019); in preprocessing, we averaged the test fMRI data across all runs for each stimulus.

The training set contains 1,200 ImageNet natural images (Deng et al., 2009), and the test set contains 50 natural images from held-out categories and 40 artificial shape images (Shen et al., 2019). We additionally evaluated a category-overlap setting to disentangle data-scale effects from dataset bias (Fig. A1). We constructed this setting by randomly selecting 50 categories from the original Deeprecon training set (150 categories × 8 images) and replacing their training images with ImageNet training images from the 50 natural-image categories used in the Deeprecon test set (8 images per category), thereby ensuring overlap for the natural-image categories. The above replacement targeted only the natural-image categories, and we did not enforce category overlap for the artificial shape test set since the notion of ImageNet category does not apply.

Our analysis mainly followed the reconstruction pipeline of Shen et al. (2019), which employs a Translator–Generator framework: a linear translator predicts latent features (VGG19) from fMRI, and a reconstruction step performs iterative image optimization to produce an image whose extracted features match the target features. To analyze ODC under a naive multivariate linear translator, we replaced the sparse translator of Shen et al. (2019) with multivariate ridge regression using a fixed penalty $\lambda = 1000$. Latent feature vectors were standardized using a standard scaler fit on the training outputs, and all projections and errors were computed in this standardized feature space.

We computed the best prediction $\hat{\boldsymbol{y}}_{\mathrm{te}}^{\mathrm{opt}}$ using Algorithm 1 with a standard scaler fit on the training outputs. To vary the data scale, we subsampled the 1,200 training images to $n \in \{600, 300, 150\}$ by selecting 4, 2, and 1 images per category (150 categories), respectively, and used one fixed subset for each $n$.

To visualize target features as images, we used Deep Image Prior (DIP) (Ulyanov et al., 2018) and optimized both the DIP network parameters and its input so that the features extracted from the generated image matched the target feature. We used AdamW (learning rate $10^{-3}$) for 800 iterations. We also report reconstructions obtained by replacing DIP with a Deep Generator Network (DGN) prior (Dosovitskiy & Brox, 2016) (Fig. A5).

Prediction error was measured as the normalized squared error $\frac{1}{d_{\mathrm{out}}}\|\hat{\boldsymbol{y}} - \boldsymbol{y}\|^2$. For comparing $\hat{\boldsymbol{y}}_{\mathrm{te}}^{\mathrm{opt}}$ and the ridge-based brain prediction $\hat{\boldsymbol{y}}_{\mathrm{te}}$, we computed the per-sample ratio $\|\hat{\boldsymbol{y}}_{\mathrm{te}}^{\mathrm{opt}} - \boldsymbol{y}_{\mathrm{te}}\|^2 / \|\hat{\boldsymbol{y}}_{\mathrm{te}} - \boldsymbol{y}_{\mathrm{te}}\|^2$ and summarized it by the mean and standard deviation across samples.

In addition, we evaluated the reconstruction method of Ozcelik & VanRullen (2023) on the same Deeprecon data. Their pipeline uses CLIP embeddings and VDVAE latents as the image representation and employs a diffusion-based generator, and we followed their original procedure without modification. The corresponding results are reported in Fig. A6.

## B.5 Proof of Corollary 1

We quantify the prediction error when the sparse regression retains a fraction $p_{\mathrm{sel}}$ of the inputs and captures a fraction $\pi$ of the truly relevant ones (Corollary 1). We focus on the asymptotic regime where both the sample size $n$ and the dimensionality $d$ are taken to infinity, while the data scale $n/d$ stays finite. By applying the risk formula for ridge regression by Dicker (2016) to the post-selection problem, we quantify the prediction error of sparse regression. In the following, $f(n,d) \approx g(n,d)$ means that $\lim_{n,d \to \infty,\, n/d \to \gamma} |f(n,d) - g(n,d)| = 0$.

*Proof.* Consider the linear model that remains after variable selection for a single coordinate $y_k$ of $\boldsymbol{y}$. Split every input vector $\boldsymbol{x}^{(i)} \in \mathbb{R}^d$ into two parts: the selected part $\boldsymbol{\xi}_k^{(i)} \in \mathbb{R}^{d_{\mathrm{sel}}}$ and the unselected part $\boldsymbol{\zeta}_k^{(i)} \in \mathbb{R}^{d-d_{\mathrm{sel}}}$, and similarly the weight vector $\boldsymbol{w}_k = W_{:,k} \in \mathbb{R}^d$ into the corresponding $\boldsymbol{u}_k \in \mathbb{R}^{d_{\mathrm{sel}}}$ and $\boldsymbol{v}_k \in \mathbb{R}^{d-d_{\mathrm{sel}}}$. With these notations

$$y_k^{(i)} = \boldsymbol{w}_k^\top \boldsymbol{x}^{(i)} + \epsilon_k^{(i)} \tag{16}$$

$$= \boldsymbol{u}_k^\top \boldsymbol{\xi}_k^{(i)} + \boldsymbol{v}_k^\top \boldsymbol{\zeta}_k^{(i)} + \epsilon_k^{(i)} \tag{17}$$

$$= \boldsymbol{u}_k^\top \boldsymbol{\xi}_k^{(i)} + \eta_k^{(i)} \tag{18}$$

where $\eta_k^{(i)} = \boldsymbol{v}_k^\top \boldsymbol{\zeta}_k^{(i)} + \epsilon_k^{(i)}$ is the combined noise term. The selected part of weight vector $\boldsymbol{u}_k$ contains $\pi s$ non-zero entries, and the unselected part $\boldsymbol{v}_k$ contains $(1-\pi)s$ non-zero entries, hence $\|\boldsymbol{u}_k\|^2 = \pi$ and $\|\boldsymbol{v}_k\|^2 = 1 - \pi$. Further, since $\boldsymbol{x}^{(i)}$ is generated from $\mathcal{N}(\boldsymbol{0}, I_d)$ and independent of $\boldsymbol{\epsilon}^{(i)}$, the combined noise satisfies

$$\eta_k^{(i)} \sim \mathcal{N}(0, \sigma_{\mathrm{eff}}^2), \tag{19}$$

with $\sigma_{\mathrm{eff}}^2 = 1 - \pi + \sigma^2$, and it is independent of $\boldsymbol{\xi}_k^{(i)}$. The post-selection problem is therefore a standard linear regression with effective dimension $d_{\mathrm{sel}}$ and effective noise variance $\sigma_{\mathrm{eff}}^2$.

The post-selection problem is an ordinary ridge regression, so an asymptotic theory of Dicker (2016) applies directly: in the limit $n, d_{\mathrm{sel}} \to \infty$ with $d_{\mathrm{sel}}/n \to \rho \in (0, \infty)$, the risk of the ridge estimator with the optimal

regularization parameter satisfies

$$\mathbb{E}\|\hat{\boldsymbol{u}}_k - \boldsymbol{u}_k\|^2 \approx \frac{\sigma_{\mathrm{eff}}^2}{2\rho}\left[\tau^2(\rho-1)-\rho+\sqrt{\left(\tau^2(\rho-1)-\rho\right)^2 + 4\rho^2\tau^2}\right], \tag{20}$$

with

$$\tau^2 = \frac{\|\boldsymbol{u}_k\|^2}{\sigma_{\mathrm{eff}}^2} = \frac{\pi}{\sigma_{\mathrm{eff}}^2}, \quad \rho = \frac{p_{\mathrm{sel}}}{\gamma}, \quad \sigma_{\mathrm{eff}}^2 = 1 - \pi + \sigma^2, \tag{21}$$

where $\gamma = n/d$ is the data scale.

For the test data $(\boldsymbol{x}_{\mathrm{te}}, \boldsymbol{y}_{\mathrm{te}})$, we can split $\boldsymbol{x}_{\mathrm{te}}$ into $\boldsymbol{\xi}_{\mathrm{te},k} \in \mathbb{R}^{d_{\mathrm{sel}}}$ and $\boldsymbol{\zeta}_{\mathrm{te},k} \in \mathbb{R}^{d-d_{\mathrm{sel}}}$ in the same way as above. The $k$-th output coordinate of the test data $\boldsymbol{y}_{\mathrm{te}}$ is similarly expressed as

$$y_{\mathrm{te},k} = \boldsymbol{u}_k^\top \boldsymbol{\xi}_{\mathrm{te},k} + \eta_{\mathrm{te},k}, \tag{22}$$

where $\eta_{\mathrm{te},k} \sim \mathcal{N}(0, \sigma_{\mathrm{eff}}^2)$ is independent of $\boldsymbol{\xi}_{\mathrm{te},k}$. The learned weight on the unselected part $\boldsymbol{\zeta}_{\mathrm{te},k}$ is zero, so the prediction for the $k$-th output coordinate is

$$\hat{y}_{\mathrm{te},k} = \hat{\boldsymbol{w}}_k^\top \boldsymbol{x}_{\mathrm{te}} \tag{23}$$
$$= \hat{\boldsymbol{u}}_k^\top \boldsymbol{\xi}_{\mathrm{te},k}, \tag{24}$$

where $\hat{\boldsymbol{u}}_k$ is the learned weight on the selected part $\boldsymbol{\xi}_{\mathrm{te},k}$. Therefore, the prediction error $R$ is

$$R = \frac{1}{d}\mathbb{E}\|\hat{\boldsymbol{y}}_{\mathrm{te}} - \boldsymbol{y}_{\mathrm{te}}\|^2 \tag{25}$$

$$= \frac{1}{d}\sum_{k=1}^{d}\mathbb{E}\left[\|\hat{y}_{\mathrm{te},k} - y_{\mathrm{te},k}\|^2\right] \tag{26}$$

$$= \frac{1}{d}\sum_{k=1}^{d}\mathbb{E}\left[\|(\hat{\boldsymbol{u}}_k - \boldsymbol{u}_k)^\top \boldsymbol{\xi}_{\mathrm{te,k}} - \eta_{\mathrm{te},k}\|^2\right] \tag{27}$$

$$= \frac{1}{d}\sum_{k=1}^{d}\mathbb{E}\left[\|(\hat{\boldsymbol{u}}_k - \boldsymbol{u}_k)^\top \boldsymbol{\xi}_{\mathrm{te,k}}\|^2 + \|\eta_{\mathrm{te},k}\|^2\right] \tag{28}$$

$$= \frac{1}{d}\sum_{k=1}^{d}\mathbb{E}\left[\|\hat{\boldsymbol{u}}_k - \boldsymbol{u}_k\|^2 + \sigma_{\mathrm{eff}}^2\right] \tag{29}$$

$$\approx \sigma_{\mathrm{eff}}^2\left\{1 + \frac{1}{2\rho}\left[\tau^2(\rho-1)-\rho+\sqrt{\left(\tau^2(\rho-1)-\rho\right)^2 + 4\rho^2\tau^2}\right]\right\}. \tag{30}$$

$$\square$$

## B.6 Correlation-based variable selection

To analyze practical sparse regression within our theoretical framework, we considered a simple correlation-based filter (akin to SIS (Fan & Lv, 2008)) as a concrete procedure (see also Fig. 7A). This choice was also motivated by practice: Shen et al. (2019) employed correlation-based voxel selection as part of their reconstruction pipeline. This method is computationally trivial compared to wrapper or embedded approaches, making it well-suited for reconstruction pipelines that must efficiently handle large-scale data. For each input vector $\mathbf{x}_j = (X_{\mathrm{tr}})_{:,j} \in \mathbb{R}^n$ and output vector $\mathbf{y}_k = (Y_{\mathrm{tr}})_{:,k} \in \mathbb{R}^n$ across the $n$ training samples, the model computes the sample Pearson correlation coefficient:

$$r_{jk} = \frac{(\mathbf{x}_j - \hat{\mu}_{\mathbf{x}_j})^\top (\mathbf{y}_k - \hat{\mu}_{\mathbf{y}_k})}{n\,\hat{\sigma}_{\mathbf{x}_j}\hat{\sigma}_{\mathbf{y}_k}},$$

where $\hat{\mu}_{\mathbf{x}_j} = \frac{1}{n}\sum_{i=1}^{n}x_j^{(i)}$ and $\hat{\mu}_{\mathbf{y}_k} = \frac{1}{n}\sum_{i=1}^{n}y_k^{(i)}$ are the sample means, and $\hat{\sigma}_{\mathbf{x}_j}$ and $\hat{\sigma}_{\mathbf{y}_k}$ are the corresponding sample standard deviations. Once the model has computed the correlations $r_{jk}$ for all input-output pairs, it

selects the top $d_{\text{sel}}$ input variables with the highest absolute correlations $|r_{jk}|$ for each output dimension $k$. A separate ridge regression uses only the selected inputs to predict each output dimension. Applying this procedure to all $k$ yields a column-wise sparse support, which may differ across outputs.

### B.7 Proof of Theorem 1

We relate the selection rate $p_{\text{sel}}$ and the hit rate $\pi$ of the correlation-based filter to characterize the performance of the practical filter (Theorem 1). We work in the same asymptotic regime as in the preceding analysis: $n, d \to \infty$ with $n/d \to \gamma \in (0, \infty)$. In the following, $f(n,d) \approx g(n,d)$ means that $\lim_{n,d\to\infty,\, n/d\to\gamma} |f(n,d) - g(n,d)| = 0$.

*Proof.* We express the column slices of data matrices $(X_{\text{tr}})_{:,j}, (Y_{\text{tr}})_{:,k} \in \mathbb{R}^n$ as $\mathbf{x}_j$ or $\mathbf{y}_k$. We intentionally use different symbol from $\boldsymbol{x}^{(i)}$ or $\boldsymbol{y}^{(i)}$ which indicates $i$-th single instances. For each output coordinate $k \in \{1, \ldots, d\}$, we can rewrite the data generation process Assumption 1 as

$$\mathbf{y}_k = X_{\text{tr}} W_{:,k} + \boldsymbol{\epsilon}_k \tag{31}$$

$$= \sum_{j=1}^{d} W_{jk} \mathbf{x}_j + \boldsymbol{\epsilon}_k, \tag{32}$$

$$= \frac{1}{\sqrt{s}} \sum_{j \in \mathcal{S}_k} \mathbf{x}_j + \boldsymbol{\epsilon}_k, \tag{33}$$

where $W_{:,k}$ is the $k$-th column of the teacher weight matrix $W$, $\boldsymbol{\epsilon}_k \in \mathbb{R}^n$ is the $k$-th output noise vector, and $\mathcal{S}_k = \{j \mid W_{jk} \neq 0\}$ is the informative index set for the $k$-th output coordinate. Within this data generation process, $\mathbf{x}_j \sim \mathcal{N}(\mathbf{0}, I_n)$ and $\boldsymbol{\epsilon}_k \sim \mathcal{N}(\mathbf{0}, \sigma^2 I_n)$ hold.

The correlation-based filter ranks each input variable $j$ according to its absolute correlation with the output $k$:

$$r_{jk} = \frac{(\mathbf{x}_j - \hat{\mu}_{\mathbf{x}_j})^\top (\mathbf{y}_k - \hat{\mu}_{\mathbf{y}_k})}{n\, \hat{\sigma}_{\mathbf{x}_j} \hat{\sigma}_{\mathbf{y}_k}} \tag{34}$$

$$= \frac{\langle \mathbf{x}_j, \mathbf{y}_k \rangle}{n\, \hat{\sigma}_{\mathbf{x}_j} \hat{\sigma}_{\mathbf{y}_k}} - \frac{\hat{\mu}_{\mathbf{x}_j} \hat{\mu}_{\mathbf{y}_k}}{\hat{\sigma}_{\mathbf{x}_j} \hat{\sigma}_{\mathbf{y}_k}} \tag{35}$$

where $\hat{\mu}_{\mathbf{x}_j} = \frac{1}{n} \sum_{i=1}^{n} x_j^{(i)}$ and $\hat{\mu}_{\mathbf{y}_k} = \frac{1}{n} \sum_{i=1}^{n} y_k^{(i)}$ are the sample means, and $\hat{\sigma}_{\mathbf{x}_j}$ and $\hat{\sigma}_{\mathbf{y}_k}$ are the corresponding sample standard deviations. Because $\mathbf{x}_j \sim \mathcal{N}(\mathbf{0}, I_n)$ and $\mathbf{y}_k \sim \mathcal{N}(\mathbf{0}, (1+\sigma^2)I_n)$ hold, the means and standard deviations satisfy

$$\hat{\mu}_{\mathbf{x}_j} \to 0, \quad \hat{\sigma}_{\mathbf{x}_j} \to 1, \quad \hat{\mu}_{\mathbf{y}_k} \to 0, \quad \hat{\sigma}_{\mathbf{y}_k} \to \sqrt{1 + \sigma^2} \qquad \text{a.s. as } n \to \infty. \tag{36}$$

The correlation coefficient satisfies

$$r_{jk} - \frac{\langle \mathbf{x}_j, \mathbf{y}_k \rangle}{n\, \hat{\sigma}_{\mathbf{x}_j} \hat{\sigma}_{\mathbf{y}_k}} \to 0 \qquad \text{a.s. as } n \to \infty \tag{37}$$

because the second term of equation 35 vanishes. Hence, thresholding $|r_{jk}|$ is asymptotically equivalent to thresholding $|z_{jk}| := |\langle \mathbf{x}_j, \mathbf{y}_k \rangle|$; thus we analyze $|z_{jk}|$ as a proxy in the following.

We first derive the expectation and variance of the inner product $z_{jk} = \langle \mathbf{x}_j, \mathbf{y}_k \rangle$ for informative inputs $(j \in \mathcal{S}_k)$ and uninformative inputs $(j \notin \mathcal{S}_k)$. For an informative input $j \in \mathcal{S}_k$, we can write

$$z_{jk} = \langle \mathbf{x}_j, \mathbf{y}_k \rangle \tag{38}$$

$$= \langle \mathbf{x}_j, \frac{1}{\sqrt{s}} \sum_{t \in \mathcal{S}_k} \mathbf{x}_t + \boldsymbol{\epsilon}_k \rangle \tag{39}$$

$$= \frac{1}{\sqrt{s}} \langle \mathbf{x}_j, \mathbf{x}_j \rangle + \frac{1}{\sqrt{s}} \sum_{t \in \mathcal{S}_k \setminus \{j\}} \langle \mathbf{x}_j, \mathbf{x}_t \rangle + \langle \mathbf{x}_j, \boldsymbol{\epsilon}_k \rangle. \tag{40}$$

Independence of the Gaussian components yields

$$\mathbb{E}[z_{jk}] = \frac{1}{\sqrt{s}}\mathbb{E}[\langle \mathbf{x}_j, \mathbf{x}_j \rangle] + \frac{1}{\sqrt{s}}\sum_{t \in \mathcal{S}_k \setminus \{j\}} \mathbb{E}[\langle \mathbf{x}_j, \mathbf{x}_t \rangle] + \mathbb{E}[\langle \mathbf{x}_j, \boldsymbol{\epsilon}_k \rangle] \tag{41}$$

$$= \frac{n}{\sqrt{s}} + 0 + 0 = \frac{n}{\sqrt{s}}, \tag{42}$$

$$\mathrm{Var}[z_{jk}] = \frac{1}{s}\mathrm{Var}[\langle \mathbf{x}_j, \mathbf{x}_j \rangle] + \frac{1}{s}\sum_{t \in \mathcal{S}_k \setminus \{j\}} \mathrm{Var}[\langle \mathbf{x}_j, \mathbf{x}_t \rangle] + \mathrm{Var}[\langle \mathbf{x}_j, \boldsymbol{\epsilon}_k \rangle] \tag{43}$$

$$= \frac{2n}{s} + \frac{(s-1)n}{s} + n\sigma^2 = n\left(\frac{s+1}{s} + \sigma^2\right). \tag{44}$$

Similarly, for an uninformative input $j \notin \mathcal{S}_k$, we find

$$z_{jk} = \langle \mathbf{x}_j, \frac{1}{\sqrt{s}}\sum_{t \in \mathcal{S}_k} \mathbf{x}_t + \boldsymbol{\epsilon}_k \rangle \tag{45}$$

$$= \frac{1}{\sqrt{s}}\sum_{t \in \mathcal{S}_k} \langle \mathbf{x}_j, \mathbf{x}_t \rangle + \langle \mathbf{x}_j, \boldsymbol{\epsilon}_k \rangle. \tag{46}$$

$$\mathbb{E}[z_{jk}] = \frac{1}{\sqrt{s}}\sum_{t \in \mathcal{S}_k} \mathbb{E}[\langle \mathbf{x}_j, \mathbf{x}_t \rangle] + \mathbb{E}[\langle \mathbf{x}_j, \boldsymbol{\epsilon}_k \rangle] \tag{47}$$

$$= 0 + 0 = 0, \tag{48}$$

$$\mathrm{Var}[z_{jk}] = \frac{1}{s}\sum_{t \in \mathcal{S}_k} \mathrm{Var}[\langle \mathbf{x}_j, \mathbf{x}_t \rangle] + \mathrm{Var}[\langle \mathbf{x}_j, \boldsymbol{\epsilon}_k \rangle] \tag{49}$$

$$= n(1 + \sigma^2). \tag{50}$$

Next, we derive the relation between the hit rate $\pi$ and the selection rate $p_{\mathrm{sel}}$ of the correlation-based filter. Fix an output $k$ and let the true support be $\mathcal{S}_k = \{j \mid w_{jk} \neq 0\}$ with $|\mathcal{S}_k| = s = ad$. For a common threshold $t_k > 0$, define the selection set $\hat{\mathcal{S}}_k(t_k) = \{j \mid |z_{jk}| > t_k\}$. Recall the selection rate $p_{\mathrm{sel}} = \frac{|\hat{\mathcal{S}}_k(t_k)|}{d} = \Pr(j \in \hat{\mathcal{S}}_k(t_k))$ and the hit rate $\pi = \Pr(j \in \hat{\mathcal{S}}_k(t) \mid j \in \mathcal{S}_k)$. Define the null retention probability $\pi_- := \Pr(j \in \hat{\mathcal{S}}_k(t_k) \mid j \notin \mathcal{S}_k)$. Noting that $a = \Pr(j \in \mathcal{S}_k)$, we obtain the identity

$$p_{\mathrm{sel}} = a\,\pi + (1-a)\,\pi_-. \tag{51}$$

We proceed by approximating $\pi$ and $\pi_-$ and then eliminating the threshold. Denote by $\Phi(\cdot)$ and $\Phi^{-1}(\cdot)$ the CDF and inverse-CDF of the standard normal distribution. Writing $u_k := t_k/\sqrt{n}$ and $\Delta := \sqrt{n/s} = \sqrt{\gamma/a}$ with data scale $\gamma = n/d$, we obtain

$$\pi = \Pr(|z_{jk}| > t_k \mid j \in \mathcal{S}_k) \tag{52}$$

$$= \Phi\left(-\frac{t_k - \frac{n}{\sqrt{s}}}{\sqrt{n\left(\frac{s+1}{s} + \sigma^2\right)}}\right) + \Phi\left(-\frac{t_k + \frac{n}{\sqrt{s}}}{\sqrt{n\left(\frac{s+1}{s} + \sigma^2\right)}}\right) + \mathcal{O}\left(n^{-\frac{1}{2}}\right) \tag{53}$$

$$= \Phi\left(-\frac{u_k - \Delta}{\sqrt{1 + \sigma^2 + \frac{1}{s}}}\right) + \Phi\left(-\frac{u_k + \Delta}{\sqrt{1 + \sigma^2 + \frac{1}{s}}}\right) + \mathcal{O}\left(n^{-\frac{1}{2}}\right) \tag{54}$$

$$= \Phi\left(-\frac{u_k - \Delta}{\sqrt{1 + \sigma^2}}\right) + \Phi\left(-\frac{u_k + \Delta}{\sqrt{1 + \sigma^2}}\right) + \mathcal{O}\left(n^{-\frac{1}{2}}\right). \tag{55}$$

Eq. 53 follows from Berry-Esseen theorem, and Eq. 55 follows from

$$\Phi\left(-\frac{u_k \pm \Delta}{\sqrt{1 + \sigma^2 + \frac{1}{s}}}\right) = \Phi\left(-\frac{u_k \pm \Delta}{\sqrt{1 + \sigma^2}}\right) + \mathcal{O}(1/s). \tag{56}$$

Similarly,

$$\pi_- = \Pr(|z_{jk}| > t_k \mid j \notin \mathcal{S}_k) \tag{57}$$

$$= 2\,\Phi\left(-\frac{t_k}{\sqrt{n(1+\sigma^2)}}\right) + \mathcal{O}\left(n^{-\frac{1}{2}}\right) \tag{58}$$

$$= 2\,\Phi\left(-\frac{u_k}{\sqrt{1+\sigma^2}}\right) + \mathcal{O}\left(n^{-\frac{1}{2}}\right). \tag{59}$$

Combining Eq. 55 and Eq. 59 with Eq. 51 yields the two-equation relation

$$p_{\text{sel}} = a\left[\Phi\left(-\frac{u_k-\Delta}{\sqrt{1+\sigma^2}}\right) + \Phi\left(-\frac{u_k+\Delta}{\sqrt{1+\sigma^2}}\right)\right]$$
$$+ (1-a)\,2\,\Phi\left(-\frac{u_k}{\sqrt{1+\sigma^2}}\right) + \mathcal{O}\left(n^{-\frac{1}{2}}\right), \tag{60}$$

$$\pi = \Phi\left(-\frac{u_k-\Delta}{\sqrt{1+\sigma^2}}\right) + \Phi\left(-\frac{u_k+\Delta}{\sqrt{1+\sigma^2}}\right) + \mathcal{O}\left(n^{-\frac{1}{2}}\right), \tag{61}$$

which links $p_{\text{sel}}$ and $\pi$ through the common threshold $u_k > 0$. Eliminating $u_k$ gives, for $a \in (0,1)$,

$$\pi \approx \Phi\left(\Phi^{-1}\left(\frac{\pi_-}{2}\right) + \alpha\right) + \Phi\left(\Phi^{-1}\left(\frac{\pi_-}{2}\right) - \alpha\right) \tag{62}$$

where

$$\pi_- = \frac{p_{\text{sel}} - a\pi}{1-a}, \quad \alpha = \frac{\Delta}{\sqrt{1+\sigma^2}} = \sqrt{\frac{\gamma}{a(1+\sigma^2)}}, \tag{63}$$

while in the boundary case $a = 1$ the relation reduces to $p_{\text{sel}} = \pi$. $\qquad\square$

### B.8 Simulation study: generative conditions and parameters

To evaluate the finite-sample accuracy of our theory and its robustness to deviations from the assumptions, we conducted simulations under the following four generative conditions:

(A) **Baseline:** Each sample pair $(\boldsymbol{x}^{(i)}, \boldsymbol{y}^{(i)})$ is generated as $\boldsymbol{y}^{(i)} = W^\top \boldsymbol{x}^{(i)} + \boldsymbol{\epsilon}^{(i)}$, where $\boldsymbol{\epsilon}^{(i)} \sim \mathcal{N}(0, \sigma^2 I)$. Each non-zero entry in the teacher matrix $W$ is set to $1/\sqrt{s}$, with the remaining entries set to 0. The input vector $\boldsymbol{x}^{(i)}$ is drawn from a standard normal distribution with independent components $\mathcal{N}(0,1)$.

(B) **Gaussian non-zero weights:** Each sample pair $(\boldsymbol{x}^{(i)}, \boldsymbol{y}^{(i)})$ is generated as in (A), but the non-zero entries in the teacher matrix are sampled from a standard normal distribution $\mathcal{N}(0, \frac{1}{s})$. Inputs $\boldsymbol{x}^{(i)}$ are independently drawn from $\mathcal{N}(0,1)$, as in (A).

(C) **Correlated signal:** Each sample pair $(\boldsymbol{x}^{(i)}, \boldsymbol{y}^{(i)})$ is generated as in (A), with the same binary non-zero structure in the teacher matrix $W$. The input vector $\boldsymbol{x}^{(i)}$ is sampled from a zero-mean multivariate normal distribution with a Toeplitz covariance matrix $\Sigma_{ij} = \rho^{|i-j|}$, where $\rho = 0.5$.

(D) **Input noise:** Each sample pair $(\boldsymbol{x}^{(i)}, \boldsymbol{y}^{(i)})$ is generated as $\boldsymbol{y}^{(i)} = W^\top \boldsymbol{z}^{(i)}$ and $\boldsymbol{x}^{(i)} = \boldsymbol{z}^{(i)} + \boldsymbol{\epsilon}^{(i)}$, where $\boldsymbol{\epsilon}^{(i)} \sim \mathcal{N}(0, \sigma^2 I)$. The teacher matrix $W$ has the same binary non-zero structure as in (A). The input vector $\boldsymbol{z}^{(i)}$ is sampled from a standard normal distribution with independent components $\mathcal{N}(0,1)$.

The simulations are performed with the following parameters: the dimensionality $d = 1000$ and the noise level $\sigma = 0.1$. For ridge regression, we used the ridge penalty that is theoretically optimal in the baseline setting, $\lambda = d/\tau^2$, in all simulation conditions (A–D).

### B.9 Noise level estimation in the Deeprecon dataset

To set the noise level used in our theoretical comparisons with real fMRI results, we estimated the trial-to-trial noise variance from repeated presentations of identical stimuli in the Deeprecon dataset (Shen et al., 2019). We used an ANOVA-like variance decomposition to separately estimate the stimulus-driven variance and the within-stimulus trial noise.

For each voxel, let $x_{i,r}$ denote the preprocessed fMRI response to stimulus $i \in \{1, \ldots, s\}$ at repetition $r \in \{1, \ldots, t\}$. We defined the per-stimulus mean and the grand mean as

$$\bar{x}_{i\cdot} = \frac{1}{t} \sum_{r=1}^{t} x_{i,r}, \qquad \bar{x}_{\cdot\cdot} = \frac{1}{st} \sum_{i=1}^{s} \sum_{r=1}^{t} x_{i,r}. \tag{64}$$

We then computed the between-stimulus and within-stimulus sums of squares,

$$\mathrm{SS}_{\text{between}} = t \sum_{i=1}^{s} (\bar{x}_{i\cdot} - \bar{x}_{\cdot\cdot})^2, \tag{65}$$

$$\mathrm{SS}_{\text{within}} = \sum_{i=1}^{s} \sum_{r=1}^{t} (x_{i,r} - \bar{x}_{i\cdot})^2, \tag{66}$$

and their corresponding mean squares,

$$\mathrm{MS}_{\text{between}} = \frac{\mathrm{SS}_{\text{between}}}{s - 1}, \qquad \mathrm{MS}_{\text{within}} = \frac{\mathrm{SS}_{\text{within}}}{s(t - 1)}. \tag{67}$$

We modeled repeated responses by a random-effects decomposition

$$x_{i,r} = \mu + z_i + \varepsilon_{i,r}, \tag{68}$$

where $z_i \sim \mathcal{N}(0, \sigma_z^2)$ captures stimulus-driven variability across stimuli and $\varepsilon_{i,r} \sim \mathcal{N}(0, \sigma_\varepsilon^2)$ captures trial-to-trial noise. Under this model, the expected mean squares satisfy $\mathbb{E}[\mathrm{MS}_{\text{within}}] = \sigma_\varepsilon^2$ and $\mathbb{E}[\mathrm{MS}_{\text{between}}] = \sigma_\varepsilon^2 + t\sigma_z^2$. We therefore estimated

$$\widehat{\sigma}_\varepsilon^2 = \mathrm{MS}_{\text{within}}, \qquad \widehat{\sigma}_z^2 = \frac{\mathrm{MS}_{\text{between}} - \mathrm{MS}_{\text{within}}}{t}. \tag{69}$$

We computed these estimates voxel-wise and then averaged $\widehat{\sigma}_\varepsilon^2 / \widehat{\sigma}_z^2$ over voxels with $\widehat{\sigma}_z^2 > 0$.

Using ImageNetTraining (1200 stimuli, 5 repetitions), we obtained $\sigma_\varepsilon / \sigma_z = \sqrt{\widehat{\sigma}_\varepsilon^2 / \widehat{\sigma}_z^2} \approx 5.7$. Using ImageNetTest (50 stimuli, 24 repetitions), we obtained $\sigma_\varepsilon / \sigma_z = \sqrt{\widehat{\sigma}_\varepsilon^2 / \widehat{\sigma}_z^2} \approx 5.2$. Overall, these estimates suggest that the noise level $\sigma_\varepsilon / \sigma_z$ was on the order of 5 and was relatively consistent across the two datasets.

### B.10 Naive vs. sparse translator comparison in the Deeprecon dataset

We compared naive and sparse translators on the Deeprecon dataset (Shen et al., 2019) using the same subject (`sub-01`), VC ROI, preprocessing, and category-balanced subsampling protocol as in the ODC analysis. We evaluated prediction error on the 50 natural test images (held-out categories). Unless stated otherwise, the target outputs were VGG19 latent features (as in Fig. 9B).

The naive translator was multivariate ridge regression with a fixed penalty $\lambda = 1000$. The sparse translator was a correlation-based filter plus ridge regression: we used a fixed ridge penalty $\lambda = 100$ and set the number of selected voxels per output dimension to $d_{\text{sel}} \in \{63, 125, 250, 500\}$ for $n \in \{150, 300, 600, 1200\}$, respectively. As a supplementary reference, we repeated the same comparison using CLIP target features (see Fig. A7). For this CLIP-target analysis, the naive translator used ridge regression with a fixed penalty $\lambda = 60000$, and the sparse translator used the same settings as above.

Prediction error was measured as the normalized squared error $\frac{1}{d_{\text{out}}} \|\hat{\boldsymbol{y}} - \boldsymbol{y}\|^2$ and averaged across the natural test stimuli.

