# OpenReview forum: "Overcoming Output Dimension Collapse: When Sparsity Enables Zero-shot Brain-to-image Reconstruction at Small Data Scales"
_TMLR — Accepted by TMLR_

### Review · Reviewer_Kipo · 2026-02-14

**Summary Of Contributions:**

This paper analyzes the translator component in "Translator-Generator" fMRI-to-image reconstruction pipelines, focusing on the low-sample regime where the number of paired fMRI-stimulus samples $n$ is smaller than the latent feature dimensionality $d_{\rm out}$ (i.e., $n/d_{\rm out}<1$). The authors compared naive multivariate linear regression against sparse linear translators as a mechanism to mitigate performance degradation under data scarcity. The core analytical observation is that, for naive linear translators, predictions are confined to the subspace spanned by training outputs, leading to an “output dimension collapse” (ODC) effect when $n/d_{\rm out}\ll 1$. Building on this, the paper proposes a best-prediction diagnostic that serves as a lower bound on achievable latent-feature error for this class of translators and is computable without brain signals. The paper further studies when sparsity can overcome ODC, providing analytical characterization in a simplified setting and empirical evidence on a benchmark dataset.

---

## Strengths

- Clear focus on a practically important bottleneck in brain-decoding pipelines, i.e., sample scarcity relative to latent dimensionality.

- The ODC characterization for naive translators is conceptually clean and yields a diagnostic that may help practitioners reason about limits induced by the chosen feature and dataset size.

- The comparison between naive and sparse linear translators provides a structured perspective on how sparsity assumptions might enable predictions beyond the training-output span.

---

## Weaknesses

- The authors sometimes phrase that the translator is "typically linear", but much recent work uses nonlinear translators (e.g. [1]). This appears to be a scoping issue: it is best to explicitly say that this paper studies linear (or linearized) translators and avoid implying that this covers the broader landscape of the research field.

- The paper is framed as “theoretical,” but several core steps rely on standard linear-algebraic properties of multivariate ridge/kernel regression. As-is, it reads more like an analytic characterization plus a stylized sparse-regression analysis. Moreover, key algebraic factors are well-known existing results. The main value seems to be in reframing and quantifying implications for brain-to-image decoding rather than introducing a new theory; the authors should position the contribution accordingly and formalize the key claims (assumptions, propositions, and precise model class).

- Some sections are difficult to follow because substantial related-work discussion is inserted into technical sections (e.g., Sections 3 and 5), disrupting the logical thread. A dedicated Related Work section would enhance clarity and minimize citation-driven detours in positioning this paper against current research.

- While sparse models can, in principle, overcome subspace restrictions, the empirical gains on realistic datasets appear modest and depend on noise levels and sparsity assumptions. The sparse-advantage argument hinges on the mapping from brain activity to features being sufficiently sparse, which may not hold for modern pretrained feature spaces (e.g., CLIP). Clarifying these assumptions and the practical scope would strengthen the paper.

[1] Scotti P S, Tripathy M, Villanueva C K T, et al. MindEye2: Shared-Subject Models Enable fMRI-To-Image With 1 Hour of Data[J]. Proceedings of Machine Learning Research, 2024, 235: 44038-44059.

**Audience:**

Yes

**Audience Explanation:**

At least part of TMLR's readers, especially those interested in representation learning and brain data modeling, would likely find this paper relevant. This work provides a clear analytical characterization of a fundamental limitation of linear translators in the low-sample, high-output-dimension regime, and proposes an interpretable diagnostic that may help quantify when a chosen feature space and dataset scale truly constrain prediction.

**Broader Impact Concerns:**

No ethical concerns were identified.

**Claims And Evidence:**

No

**Claims Explanation:**

The claims are partially supported, but key conclusions require important qualifications. Although the analytical results for the naive linear translator are clear, the empirical evidence for the practical advantage of the sparse linear translator is less convincing: on real datasets, the improvements appear modest (especially in the low-sample regime, as in Fig. 9), and the proposed explanation relies on assumptions about noise levels and "brain-like" structure that may not hold for comment pretrained feature spaces, such as CLIP features. In addition, several statements seem at risk of being mis-scoped by implying translators are typically linear, whereas many recent fMRI-to-image pipelines employ nonlinear translators.

**Requested Changes:**

All suggested changes are included in the "Weaknesses" section of the review summary.

---

> ### Author Response · Authors · 2026-03-04
> **Response to Reviewer Kipo**
>
> Thank you for the thoughtful and constructive feedback. We revised the manuscript to clarify the scope, adjust the framing to avoid overclaiming, and improve readability. We respond to your points below.
>
> ### (1) Scope of “the translator is typically linear”
>
> Our analysis is restricted to linear translators; nonlinear translators are outside the scope of this paper. We removed wording that could be read as claiming linearity is “typical,” and we added explicit scope statements in the Introduction and Preliminaries.
>
> ### (2) “Theoretical” framing and what is actually new
>
> We agree that our previous “theoretical” framing was not appropriate: Sec. 3 (now Sec. 4) relies on standard linear-algebraic properties of multivariate linear regression, and our previous framing could be interpreted as presenting these as a new theory. The main value lies in reframing and quantifying their implications for brain-to-image decoding, as you pointed out. This aligns with Reviewer YCsr’s Critical-1 (sharpen novelty beyond an “obvious” algebraic observation) and Reviewer kVtt’s Critical-1 (avoid overclaiming Contribution 1).
>
> Accordingly, we revised the Abstract and Introduction to describe the paper as a mathematical analysis of translator behavior under small data scales in reconstruction pipelines, and we revised Sec. 3 (4) to present the output-span restriction as a standard property while highlighting its practical implications for zero-shot reconstruction.
>
> To clarify what is theoretically new, we also reorganized the later sections: results that directly follow existing theory are marked accordingly (e.g., as corollaries), while the new parts are stated as theorem-style results.
>
> ### (3) Related work mixed into technical sections
>
> Following your suggestion, we created a dedicated Related work section and consolidated the related-work discussion there. In addition (responding also to Reviewer kVtt’s Strengthen-4 on brevity), we shortened Sections 3 and 4 by deleting redundant exposition. Together, these changes allow the technical sections to focus on definitions, claims, and derivations without interruption.
>
> ### (4) Assumptions and practical scope of sparse advantage
>
> We also added experiments for the sparse-regression analysis using CLIP features as an additional latent space (Fig. A7), as recommended by Reviewer YCsr (“Broaden experimental coverage”)**.** Sparse regression does not outperform naive regression with CLIP target features. CLIP is generally considered a more distributed representation and thus is expected to exhibit weaker sparsity; we clarify this point in the Discussion.
>
> The practical advantage of sparse linear translators is conditional and can be modest on current datasets. Our goal is not to claim broad practical improvements, but to characterize when sparsity helps and when it does not. To clarify assumptions and practical scope, we added a sensitivity analysis (Fig. 9C). This also addresses Reviewer YCsr’s Strengthen-6 to explain why the real-data sparsity benefit can be modest via a sensitivity-style analysis.
>
> ---
>
> We believe these revisions clarify the scope, novelty, and prerequisites of the paper, and better align our claims with the presented evidence. We sincerely thank the reviewer again for the constructive feedback.

---

### Review · Reviewer_YCsr · 2026-02-18

**Summary Of Contributions:**

This paper studies a practical bottleneck in modern brain-to-image reconstruction pipelines: the translator that maps brain activity to high-dimensional latent image features under small data scales (few paired brain–image samples relative to latent dimensionality). The authors formalize and analyze a phenomenon called output dimension collapse (ODC) for naive multivariate linear regression translators (e.g., ridge/kernel linear/PLS), showing that predictions are constrained to the linear span of the training outputs, yielding an intrinsic generalization limitation when $\(n \ll d_{\mathrm{out}}\)$ They derive an explicit expression for the best achievable prediction under the collapsed subspace (an orthogonal projection), and use it as a diagnostic lower bound to quantify how much error is fundamentally irreducible for naive translators.

The paper then argues that sparse multivariate regression (with per-output variable selection) is not structurally forced into the same output-span restriction, and develops a student–teacher analysis that predicts test error as a function of data scale, sparsity (non-zero weight ratio), and variable selection quality (selection rate + hit rate). A concrete correlation-based filter is analyzed to connect these quantities and generate error predictions, with simulations and a small empirical comparison on Deeprecon suggesting that on current fMRI data, gains from sparsity are present but modest due to noise.

**Strengths**
1. Clear identification and formalization of an important failure mode (ODC) for common linear translators.

2. The “best prediction under collapse” projection is a useful diagnostic that depends only on training/test outputs and can be computed pre-experiment.

3. Solid theoretical development for sparse regression in the student–teacher setting; connects hit rate/selection rate to expected error, and provides interpretable guidance.

4. Empirical illustrations (reconstruction degradation as $n$ shrinks; best-prediction vs brain-prediction proximity) help ground the theory.

**Weaknesses / limitations**
1. The key “ODC” claim for naive methods is mathematically true but also close to a known representer-style property for many linear multi-output regressors; the paper’s novelty hinges on how convincingly it links that property to practical zero-shot reconstruction failure beyond dataset bias, and how broadly the empirical evidence supports that link.

2. The empirical section is somewhat narrow (primarily Deeprecon + DIP-based inversion protocol + one additional baseline). It is not fully clear how robust conclusions are across latent choices (CLIP vs CNN layers), generator types (diffusion vs DIP), preprocessing, and subjects.

3. The student–teacher model assumptions (Gaussian inputs, simplified sparse teacher) are reasonable for tractability, but the mapping from real brain feature geometry (correlations, structured noise, voxel dependencies) to model parameters (e.g., sparsity ratio
$a$, noise $σ$) is only partially validated.

**Additional Comments:**

1. I appreciated the separation between within-subspace vs out-of-subspace error and the argument that the latter is irreducible for naive translators at small data scales; that decomposition gives a clean mental model for why reconstructions blur even before considering brain noise.

2. The discussion around generators “recovering” plausible images even from perturbed latents is important; it would be helpful to more explicitly tie this to evaluation practice (e.g., why latent error metrics alone may be insufficient, and when recovery checks are necessary).

3. **Minor clarity**: please ensure consistent notation between $\d_{\mathrm{in}}\$, $\d_{\mathrm{out}}\$, and the student–teacher section where $\d_{\mathrm{in}}\$ = $\d_{\mathrm{out}}\$ =$ \d_{\mathrm{}}\$ is assumed for simplicity.

**Audience:**

Yes

**Audience Explanation:**

TMLR readers working on representation learning, statistical learning under high-dimensional regimes, and neuro-AI interfaces will likely care about this paper’s message: many widely used linear translators in brain decoding have a structural generalization bottleneck when $\(n \ll d_{\mathrm{out}}\)$ , and sparsity/variable selection provides a principled route to improved sample efficiency when the mapping is appropriately sparse. The “best prediction under collapse” is also a neat diagnostic tool that can inform experimental design and evaluation for reconstruction pipelines.

**Broader Impact Concerns:**

This work is primarily theoretical/diagnostic and seems aimed at improving scientific validity of brain-to-image decoding under limited data, which is a positive contribution. That said, brain decoding and image reconstruction can raise privacy and misuse concerns if deployed outside research settings (e.g., attempts to infer private visual experiences without consent). While the paper focuses on fMRI datasets and highlights limitations (noise, irreducible error, and risks of generators hallucinating content), I recommend adding a brief Broader Impact note emphasizing:

(i) reconstruction is not “mind reading” and is constrained by stimulus sets, model assumptions, and noise;

(ii) any future applications should require informed consent and strong governance; and

(iii) diffusion/generative priors can introduce misleading details, so claims about recovered perceptual content should be conservative.

**Claims And Evidence:**

Yes

**Claims Explanation:**

The algebraic claims about output-span restriction for ridge/kernel linear/PLS and the projection-form best predictor are clear and convincing. The theoretical sparse-regression error analysis is reasonably well-motivated and supported by simulations under several deviations from the idealized assumptions. The main remaining gap is empirical breadth: the practical significance of “small data scale alone induces ODC” is supported on the chosen dataset/protocol, but would be more convincing with additional analyses that vary $\(d_{\mathrm{out}}\)$ out , latent representations, and generator choices to confirm the claimed mechanism and its generality across reconstruction settings.

**Requested Changes:**

**Critical to recommend acceptance**
1. **Clarify the precise novelty vs. “predictions lie in the span of training outputs”**: The span restriction is derived cleanly (ridge, kernel linear, PLS), but the paper should more explicitly position what is new: (i) framing it as ODC at small data scales (not just dataset bias), (ii) the best-prediction projection diagnostic, and (iii) the sparse-regression error characterization tailored to translator design. Right now, some readers may feel the core algebraic observation is “obvious,” so the manuscript needs sharper articulation of the conceptual and practical step beyond that observation.

2. **Strengthen empirical validation that small-data-scale ODC (not only dataset bias) is a dominant mechanism**: The current experiments show that the best prediction error increases as $\(n\)$ decreases, and that best prediction error explains a substantial portion of brain prediction error for naive translators. I’d like to see at least one additional analysis that isolates small data scale from other confounds—e.g., varying latent dimensionality $\(d_{\mathrm{out}}\)$ (or using controlled dimensionality reduction) at fixed
$\(n\)$ showing the predicted collapse effect scales with $\(\frac{n}{d_{\mathrm{out}}}\)$ in practice.

3. **Make the “best prediction” diagnostic more actionable and reproducible**: Since it’s a key contribution, please provide (in main text or clearly referenced appendix) a short practical recipe: what matrices are needed, how to compute the projection stably at large $\(d_{\mathrm{out}}\)$, and what computational cost to expect (especially when $\(d_{\mathrm{out}} \sim 10^{6}\)$. Also clarify any normalization/scaling choices used in the experiments.

**Would strengthen the work (non-critical but recommended)**

4. **Broaden experimental coverage**:
* Add at least one latent space beyond the one used (e.g., different CNN layers or CLIP embedding) to support claims about “brain-like” sparsity and when sparse translators should help.
* Consider a generator swap (e.g., demonstrate whether the gap between true-feature reconstructions and best-prediction reconstructions persists under a diffusion generator, even if image fidelity is high). This would help connect latent-feature error to perceptual reconstruction outcomes and to the “recovery check” discussion.
5. **Quantify uncertainty and variability**: Provide confidence intervals / variability over subjects (if possible), runs, and choices of regularization. The current plots are suggestive, but more statistical grounding would make the empirical conclusions more convincing.

6. **Explain the real-data “modest sparsity benefit” more crisply**: The paper attributes modest gains to high noise and limited sparsity; consider adding a short sensitivity analysis showing how the predicted advantage changes with $σ$ and plausible $a$, and where current datasets likely sit on that map.

---

> ### Author Response · Authors · 2026-03-04
> **Response to Reviewer YCsr (1)**
>
> Thank you for the detailed and insightful review. We particularly appreciate the point that the paper’s value depends not on stating the mathematical property itself, but on demonstrating its implications for brain-to-image reconstruction. We revised the manuscript to strengthen empirical evidence, make the best-prediction diagnostic actionable, broaden experimental coverage, and clarify when sparsity helps. We respond to your points below.
>
> ### (1) Clarify the precise novelty
>
> We agree that the output-span restriction itself can feel “obvious” to some readers, and that the novelty hinges on the conceptual and practical step beyond that algebraic fact. This aligns with Reviewer Kipo’s Critical-1, which emphasizes that the main value lies in reframing and quantifying implications for brain-to-image decoding rather than introducing a new theory, and with Reviewer kVtt’s Critical-1, which notes that our previous wording could be read as overstating theoretical novelty.
>
> Accordingly, we revised the Abstract and Introduction to describe the paper as a mathematical analysis of translator behavior under small data scales in reconstruction pipelines, and we rewrote the contribution statement to more precisely reflect our contributions. In addition, as noted in Reviewer kVtt’s Critical-2, our previous wording in Sec. 3 (now Sec. 4) could be interpreted as presenting a new theory. We revised this section to present the output-span restriction as a standard property of multivariate linear regression, while emphasizing its practical implications for zero-shot reconstruction.
>
> ### (2) Strengthen empirical validation that small-data-scale ODC (not only dataset bias) is a dominant mechanism
>
> To reduce confounds from category mismatch, we added an analysis where train/test categories are intentionally overlapped. Even in this setting, ODC-related errors and the explanatory power of the best-prediction diagnostic worsen as data scale decreases (Fig. A1), supporting that limited data scale itself can drive collapse.
>
> ### (3) Make the best-prediction diagnostic more actionable
>
> We added an implementable recipe for computing best prediction, including pseudocode, computational cost, and scaling/normalization details (Appendix B2), together with practical notes at large output dimensionality.
>
> ### (4) Broaden experimental coverage
>
> #### (4a) Additional latent space(s)
>
> The analysis that varies the latent space for ODC corresponds to the results already reported in Fig. A6, where we use CLIP and VDVAE features as the latent representation. In this setting, the irreducible error accounts for most of the total error.
>
> We also added experiments for the sparse-regression analysis using CLIP features as an additional latent space (Fig. A7). Sparse regression does not outperform naive regression with CLIP target feature. CLIP is a more distributed representation and thus is expected to exhibit weaker sparsity; we clarify this point in the Discussion. This also addresses Reviewer Kipo’s W4 concern that the sparsity assumption may not hold for more distributed feature spaces such as CLIP.
>
> #### (4b) Generator swap
>
> We agree that the strength of the generator can affect the final reconstruction quality. To address this point, we added reconstructions using a stronger image prior by replacing the DIP prior with a Deep Generator Network (DGN) prior (Fig. A5).
>
> We note that the reconstruction method used in our main figures is an inversion-style pipeline that generates images from encoder feature representations; therefore, it is difficult to combine the *same* latent features with a diffusion generator. For diffusion-based generation, the results necessarily involve a different latent representation; we report such results in Fig. A6.  We revised to refer to this figure to inform and motivate the recovery-check discussion.
>
> ### (5) Quantify uncertainty and variability
>
> To address variability across subjects, we added results for additional Deeprecon subjects (Figs. A3–A4).
>
> Regarding runs, we follow the standard Deeprecon evaluation protocol (Shen et al. 2019), where test fMRI responses are averaged across runs for each stimulus; thus, run-to-run variability is not directly available under this evaluation setup.
>
> Regarding the ridge penalty λ, using a better-tuned λ would typically reduce the total prediction error of the translator, which in turn makes the fraction explained by the best-prediction larger. Our reported proportion attributable to the best-prediction error is therefore not an overestimate resulting from a suboptimal choice of λ. For this reason, we do not expect variability across ridge-penalty choices to affect our main conclusion.

---

> ### Author Response · Authors · 2026-03-04
> **Response to Reviewer YCsr (2)**
>
> ### (6) Explain the real-data “modest sparsity benefit” more crisply
>
> A closely related point is also raised in Reviewer Kipo’s W4. We added a sensitivity map that summarizes when sparse translators are expected to help (Fig. 9C), and we locate the real-data setting on this map to explain why gains can be modest under current noise levels and sparsity.
>
> ### (7) Broader impact
>
> We added a brief Broader Impact paragraph discussing privacy/consent considerations and the need for conservative interpretation of reconstructions.
>
> ### (8) Notation consistency between $d_\mathrm{in}$, $d_\mathrm{out}$, and $d$
>
> We revised the manuscript to use $d_\mathrm{in}$ and $d_\mathrm{out}$ in the general setting and to explicitly define
>
> $d:=d_\mathrm{in}=d_\mathrm{out}$ in the student–teacher analysis.
>
> ---
>
> We believe these revisions clarify the novelty of the paper and provide stronger evidence supporting our claims. We sincerely thank the reviewer again for the constructive feedback.

---

### Review · Reviewer_kVtt · 2026-02-19

**Summary Of Contributions:**

This paper studies data scarcity in brain-to-image reconstruction through the lens of sparse linear regression. Standard linear regression is known to produce predictions which lie in the span of the training outputs, thus leading to poor performance when data is scarce compared to the feature dimension (termed “output dimension collapse” or ODC by previous work). This paper assesses how ODC affects reconstruction by comparing to the best possible prediction in the subspace spanned by the training outputs. Then, sparse linear regression is proposed as a method to avoid ODC. Theoretical analysis in a Gaussian teacher-student setting supports this claim, as do experiments in a variety of synthetic and benchmark dataset reconstructions.

Summary of strengths:
1. The insight that sparsity is sufficient to rectify ODC in brain-to-image reconstruction could be valuable for future research.
2. The experiments are comprehensive across a variety of synthetic and benchmark datasets, and their results clearly support the findings of the paper.
3. The theoretical results are clearly communicated and notation is explained well.
4. The paper is well-written (if somewhat lengthy for the content).

Summary of weaknesses:
1. The actual contributions of this paper are misaligned with what the introduction claims: the first contribution is overclaimed, while the third appears to be the main contribution of the work but is not communicated as such.
2. Certain theoretical statements in Section 3 are misleading.

**Additional Comments:**

1. In the high-dimensional linear regression literature, it is more common to use $\asymp$ instead of $\approx$ for asymptotic equality.
2. It would be great to be more specific about necessary conditions on $n/d$. Do the theoretical results allow for $n/d\to 0$, i.e., the heavily overparameterized regime? Must we have $n/d = c$ or is $n/d\to c$ allowed, for a constant $c<1$?
3. Please include x-axis ticks in Figure 3a.

**Audience:**

Yes

**Audience Explanation:**

As previously mentioned, the fact that linear regression experiences ODC at small data scales is not too interesting due to the well-known fact that its solution lies in the span of the data. However, the insight that sparsity can rectify ODC in brain-to-image reconstruction, and the theoretical analysis of the proposed correlation-based filter method, could be quite valuable.

**Claims And Evidence:**

No

**Claims Explanation:**

The contributions of this paper as stated in the abstract and introduction are somewhat overclaimed, or at least it is unclear what this paper contributes vs what is standard in the regression literature. I assess each contribution individually below.
1. “a theoretical demonstration that naive multivariate linear regression exhibits output dimension collapse at small data scales.” This does not reflect a contribution of this paper, but rather an immediate consequence of well-known properties of linear regression. The relevant Section 3 contains some misleading theoretical analysis, as exemplified by the following sentence: “Importantly, this property is not the familiar one that confines predictions to the span of the training inputs, but instead that it confines predictions to the span of the training outputs.” In linear (ridge) regression, it is well-known that the **weights** lie in the span of the training **inputs**, while the **predictions** lie in the span of the training **outputs**. Concretely, the columns of $\hat{W}$ lie in the span of $\\{\textbf{x}^{(i)}\\}\_{i=1}^n$, and the prediction $\hat{\textbf{y}}\_{\textnormal{te}}$ lies in the span of $\\{\textbf{y}^{(i)}\\}\_{i=1}^n$. One can see these are algebraically equivalent by writing the estimator in the dual form as in Eq 3-5, and this is a standard fact.
2. “an analytic diagnostic of the practical impact of ODC via the best prediction under collapse”. This contribution is accurate and well-supported by experiments.
3. “a characterization of when sparsity enables accurate prediction at small data scales through quantitative error predictions for naive and sparse translators”. This seems to me the main contribution of the work, but I only came to that understanding after re-reading the paper. The messaging is somewhat muddled by the emphasis placed on Sections 3 and 4, which primarily serve to set the stage for the investigation into sparsity and do not constitute the main contributions of the work. I reviewed the theoretical analysis in Appendices B.4 and B.6 and it appears accurate; the experiments are similarly comprehensive and support the conclusions of Sections 5 and 6.

**Requested Changes:**

Critical for acceptance:
1. A substantial revision of the writing/positioning of this paper to avoid claiming Section 3 results as a contribution and emphasizing sparsity as the main object of study. Consider splitting the third contribution “a characterization of when sparsity enables accurate prediction at small data scales through quantitative error predictions for naive and sparse translators” into two, and highlight that it is the main focus of the work.
2. Correct the previously mentioned misleading theoretical statements in Section 3.
3. Clarify in the paper which theoretical results are contributions of this work, and which are from the literature. Specifically, Section 3 (with Appendix B.1 and B.2) appears to be standard, Section 5 (with Appendix B.4) is a corollary of Dicker (2016), and Section 6 (with Appendix B.6) is the main theoretical contribution of this work. It would help to delineate these more clearly with amsthm environments, e.g., putting Eq 12-13 in a \corollary or \lemma environment and Eq 14-15 in a \theorem environment.

Would strengthen the work:
1. The paper would benefit from greater brevity, especially in Sections 3, 4, and 7.

---

> ### Author Response · Authors · 2026-03-04
> **Response to Reviewer kVtt (1)**
>
> Thank you for the detailed comments. In particular, we take your point seriously that the description in Sec. 3 (now Sec. 4) could give the impression that it presents a “new theoretical result,” which would make our contribution claims look overstated. Based on your feedback, we (i) substantially revised the positioning and wording of Section 3 and (ii) revised the manuscript to clarify what is already known and what is the added value of this work. We respond to your points below.
>
> ### (1) Writing/positioning to avoid claiming Section 3 results as a contribution and emphasizing sparsity as the main object of study
>
> In the previous version, the paper could be read as a theoretical paper (e.g., phrasing such as “we provide a theoretical analysis”). Although Sec. 3 (now Sec. 4) relies on standard linear-algebraic properties of multivariate linear regression, our wording in this section could be interpreted as presenting a new theory. We recognize this as an important issue.
>
> At the same time, we consider that the contribution of this paper is not only the sparse-regression theory. The contribution of Sections 3–4 in the previous version (now Sections 4–5) is that, **in the context of brain-to-image reconstruction**, we characterize this standard property as a practical failure mode under small data scales and formulate it in a diagnostically testable way. The value of this reconstruction-facing contribution was also recognized by the other reviewers (Reviewer Kipo’s S1/S2 and Reviewer YCsr’s S1/S2). That said, our previous writing did not state this contribution sharply enough, which could understandably make this part look like a restatement of an “obvious” algebraic fact. Indeed, Reviewer Kipo’s W2 noted that “the main value seems to be in reframing and quantifying implications for brain-to-image decoding,” and Reviewer YCsr’s W1 remarked that some readers may find the core algebraic observation “obvious,” calling for a sharper articulation of the conceptual and practical step beyond that observation.
>
> Accordingly, we revised the Abstract and Introduction to describe the paper as a mathematical analysis of translator behavior under small data scales in reconstruction pipelines (rather than presenting it as a “theoretical paper” in that sense). We also revised the contribution statement by merging the previous Contribution 1 into Contribution 2 and rewriting it so that it cannot be read as a standalone theoretical contribution.
>
> We agree that the sparse-regression results should be emphasized more clearly. Following your suggestion, we split the previous Contribution 3 into two items.
>
> ### (2) Revise misleading theoretical statements in Section 3
>
> We agree that our previous wording in Sec. 3 (now Sec. 4) was problematic. We revised this section to remove all misleading theoretical statements and to present the output-span restriction as a standard property of multivariate linear regression, while emphasizing its practical implications for zero-shot reconstruction.
>
> ### (3) Clarify which theoretical results are contributions of this work
>
> As noted above, we now state explicitly that Sec. 3 (now Sec. 4) is a standard property of multivariate linear regression. In addition, following your suggestion, we clarified the status of the later theoretical results by presenting Sec. 5 (now Sec. 6) as a corollary and Sec. 6 (now Sec. 7) as a theorem.
>
> ### (4) Improve brevity, especially around Sections 3/4/7
>
> As discussed in (1), we believe that Sections 3/4 (now Sections 4/5) are an important part of the paper because they characterize ODC in the brain-to-image reconstruction context. At the same time, we agree that the manuscript was too long and could become hard to read. To improve brevity, we removed redundant exposition and relocated related-work discussions to a dedicated Related Work section (also in response to Reviewer Kipo’s W3). These changes make the text more concise and allow the technical sections to focus on definitions, claims, and derivations without interruption.

---

> ### Author Response · Authors · 2026-03-04
> **Response to Reviewer kVtt (2)**
>
> ### (5) Additional comments
>
> #### **(5a) Notation for asymptotic equality.**
>
> In the previous version, we did not explicitly define our notion of asymptotic equality. In the high-dimensional linear regression literature, we understand that $\asymp$ is commonly used to denote equivalence up to constant factors. Since our notion is different, we use $\approx$ and define it explicitly:$f(n,d) \approx g(n,d)$means
>
> $\lim_{n,d\to\infty, n/d\to\gamma} \bigl|f(n,d)-g(n,d)\bigr| = 0.$
>
> At the same time, we revised places where $\approx$ had been used in a way that could be interpreted as convergence in probability.
>
> #### **(5b) Asymptotic regime.**
>
> We work in the asymptotic regime $n,d \to \infty$ with $n/d \to \gamma \in (0,\infty)$. In particular, we do not consider the regime $n/d \to 0$, while convergence to a positive constant $n/d \to c$ is allowed. We revised the manuscript to state this asymptotic regime explicitly.
>
> #### **(5c) Figure readability.**
>
> We added x-axis ticks in Fig. 3a so that the corresponding sample sizes are clearly visible.
>
> ---
>
> Thank you again for your review. Your comments were very helpful not only for clarifying the positioning of our contributions, but also for improving the theoretical precision of the manuscript.

---

### Author Response · Authors · 2026-03-04
**General Response**

Thank you very much for taking the time to review our manuscript. We have revised the paper accordingly and would appreciate your further consideration. Changes in the manuscript are highlighted in blue.

Major updates in this revision include:

- Repositioning the manuscript to avoid overclaiming theoretical novelty, and revising the ODC section to state the output-span restriction as a standard property while clarifying its implications for zero-shot reconstruction.
- Making the boundary between standard results and our contributions explicit (including theorem-style organization for the later theoretical results).
- Making the best-prediction diagnostic directly reproducible by adding pseudocode, computational cost, and scaling/normalization details.
- Improving readability by separating Related Work and streamlining technical exposition.
- Strengthening empirical support and clarifying practical scope (additional validations, subject variability, and a sensitivity analysis explaining when sparsity helps and why gains can be modest).

We respond to each reviewer in their respective review thread.

---

### Decision · Action_Editor_ZnmJ · 2026-03-23

**Recommendation:** Accept as is

**Audience:**

Yes

**Audience Explanation:**

Yes. I believe at least part of TMLR’s audience would be interested in this paper. The work will be particularly relevant to readers interested in high-dimensional regression, representation learning under data scarcity, neuro-AI interfaces, and diagnostic analysis of model bottlenecks in structured prediction pipelines. Beyond the specific brain-to-image reconstruction setting, the paper highlights a useful and general perspective on how output-space geometry and sample scale can create structural limitations for linear translators. The best-prediction diagnostic and the analysis of when sparsity can mitigate this limitation are likely to be of interest not only to researchers in brain decoding, but also to those studying generalization in low-sample, high-dimensional regimes.

**Claims And Evidence:**

Yes

**Claims Explanation:**

The claims made in the revised submission are, on the whole, supported by clear and convincing evidence. The paper studies an important bottleneck in brain-to-image reconstruction pipelines under small data scales, and the revised version now positions its contributions more carefully: it no longer overstates the standard output-span property of naive multivariate linear regression as a novel theoretical result, and it more clearly distinguishes between known regression facts, the paper’s practical diagnostic contribution, and the new sparse-regression analysis. The mathematical development is coherent, and the empirical results on synthetic settings and benchmark reconstruction data are generally aligned with the paper’s main conclusions. In particular, the best-prediction diagnostic is now made more actionable and reproducible, and the additional experiments and discussion help clarify when sparsity is expected to help and why the benefit may be modest on current noisy datasets. While the empirical scope remains somewhat specialized and the practical gains of sparse regression are not dramatic in current real-data regimes, the paper’s claims are now measured and appropriately qualified. Overall, I find the revised manuscript technically sound and supported by adequate evidence for publication in TMLR.